# Analysing the Update step in Graph Neural Networks via Sparsification

## Abstract

In recent years, Message-Passing Neural Networks (MPNNs), the most prominent Graph Neural Network (GNN) framework, have celebrated much success in the analysis of graph-structured data. In MPNNs the computations are split into three steps, Aggregation, Update and Readout. In this paper a series of models to successively sparsify the linear transform in the Update step is proposed. Specifically, the ExpanderGNN model with a tuneable sparsification rate and the Activation-Only GNN, which has no linear transform in the Update step, are proposed. In agreement with a growing trend in the relevant literature the sparsification paradigm is changed by initialising sparse neural network architectures rather than expensively sparsifying already trained architectures. These novel benchmark models enable a better understanding of the influence of the Update step on model performance and outperform existing simplified benchmark models such as the Simple Graph Convolution (SGC). The ExpanderGNNs, and in some cases the Activation-Only models, achieve performance on par with their vanilla counterparts on several down-stream graph prediction tasks, while containing exponentially fewer trainable parameters. In experiments with matching parameter numbers our benchmark models outperform the state-of-the-art GNNs models. These observations enable us to conclude that in practice the update step often makes no positive contribution to the model performance.

## 1 Introduction

Recent years have witnessed the blossom of Graph Neural Networks (GNNs). They have become the standard tools for analysing and learning graph-structured data (Wu et al., 2020) and have demonstrated convincing performance in various application areas, including chemistry (Duvenaud et al., 2015), social networks (Monti et al., 2019), natural language processing (Yao et al., 2019) and neural science (Griffa et al., 2017).

Among various GNN models, Message-Passing Neural Networks (MPNNs, Gilmer et al. (2017)) and their variants are considered to be the dominating class. In MPNNs, the learning procedure can be separated into three major steps: *Aggregation*, *Update* and *Readout*, where *Aggregation* and *Update* are repeated iteratively so that each node's representation is updated recursively based on the transformed information aggregated over its neighbourhood. With each iteration, the receptive field of the hidden representation is increased by 1-step on the graph structure such that at $k^{\text{th}}$ iteration, the hidden state of node $i$ is composed of information from its $k$-hop neighbourhood.

There is thus a division of labour between the *Aggregation* and the *Update* step, where the *Aggregation* utilises local graph structure, while the *Update* step is only applied to single node representations at a time independent of the local graph structure. From this a natural question then arises: What is the impact of the graph-agnostic *Update* step on the performance of GNNs?

Wu et al. (2019) first challenged the role of *Update* steps by proposing a simplified graph convolutional network (SGC) where they removed the non-linearities in the *Update* steps and collapsed the consecutive linear transforms into a single transform. Their experiments demonstrated, surprisingly, that in some instances the *Update* step of Graph Convolutional Network (GCN, Kipf & Welling (2017)) can be left out completely without the models' accuracy decreasing.

In the same spirit, we propose in this paper to analyse the impact of the *Update* step in a systematic way. To this end, we propose two nested model classes, where the *Update* step is successively sparsified. In the first model class which we refer to as *Expander* GNN, the linear transform layers of the *Update* step are sparsified; while in the second model class, the linear transform layers are removed and only the activation functions remain in the model. We name the second model *Activation-Only* GNN and it contrasts the SGC where the activation functions where removed to merge the linear layers.

Inspired by the recent advances in the literature of sparse Convolutional Neural Network (CNN) architectures (Prabhu et al., 2018), we propose to utilise expander graphs as the sparsifier of the linear layers (hence the model's name). Guided by positive graph theoretic properties, it optimises sparse network architectures at initialisation and accordingly saves the cost of traditional methods of iteratively pruning connections during training.

Through a series of empirical assessments on different graph learning tasks (graph and node classification as well as graph regression), we demonstrate that the *Update* step can be heavily simplified without inhibiting performance or relevant model expressivity. Our findings partly agree with the work in (Wu et al., 2019), in that dense *Update* steps in GNN are expensive and often unnecessary. In contrast to their proposition, we find that there are many instances in which leaving the *Update* step out completely significantly harms performance. In these instances our *Activation-Only* model shows superior performance while matching the number of parameters and efficiency of the SGC.

Our contributions can be summarised as follows.

(1) We explore the impact of the *Update* step in MPNNs through the newly proposed model class of *Expander* GNNs with tuneable density. We show empirically that *a sparse update step matches the performance of the standard model architectures*.
(2) As an extreme case of the *Expander* GNN, as well as an alternative to the SGC, we propose the *Activation-Only* GNNs that remove the linear transformation layer from the *Update* step and keep non-linearity in tact. We observe the *Activation-Only* models to exhibit comparable, sometimes significantly superior performance to SGC while being equally time and memory efficient.

Both of our proposed model classes can be extrapolated without further efforts to a variety of models in the MPNN framework and hence provide practitioners with an array of efficient and often highly performant GNN benchmark models.

The rest of this paper is organised as follows. In Section 2, we provide an overview of the related work. Section 3 introduces preliminary concepts of MPNNs and expander graphs, followed by a detailed presentation of our two proposed model classes. Section 4 discusses our experimental setting and empirical evaluation of the proposed models in a variety of downstream graph learning tasks.

## 2 RELATED WORKS

In recent years the idea of *utilising expander graphs in the design of neural networks* is starting to be explored in the CNN literature. Most notably, Prabhu et al. (2018) propose to replace linear fully connected layers in deep networks using an expander graph sampling mechanism and hence, propose a novel CNN architecture they call X-nets. The great innovation of this approach is that well-performing sparse neural network architectures are initialised rather than expensively calculated. Furthermore, they are shown to compare favourably in training speed, accuracy and performance trade-offs to several other state of the art architectures. McDonald & Shokoufandeh (2019) and Kepner & Robinett (2019) build on the X-net design and propose alternative expander sampling mechansions to extend the simplistic design chosen in the X-nets. Independent of this literature branch, Bourely et al. (2017) explore 6 different mechanisms to randomly sample expander graph layers. Across the literature the results based on expander graph layers are encouraging.

Recently, two papers observed that *simplifications in the update step of the GCN model* is a promising area of research (Wu et al., 2019; Salha et al., 2019). Wu et al. (2019) proposed the Simple Graph Convolution (SGC) model, where simplification is achieved by removing the non-linear activation functions from the GCN model. This removal allows them to merge all linear transformations in the update steps into a single linear transformation without sacrificing expressive power. Salha et al. (2019) followed a similar rationale in their simplification of the graph autoencoder and variational

graph autoencoder models. These works have had an immediate impact on the literature featuring as a benchmark model and object of study in many recent papers. The idea of omitting update steps guided Chen et al. (2020) in the design of simplified models and has found successful application in various areas where model complexity needs to be reduced (Waradpande et al., 2020; He et al., 2020) or very large graphs ($\sim 10^6$ nodes/edges) need to be processed (Salha et al., 2020).

In our work we aim to extend these efforts by providing more simplified benchmark models for GNNs without a specific focus on the GCN.

## 3 INVESTIGATING THE ROLE OF THE UPDATE STEP

In this section, we present the two proposed model classes, where we sparsify or remove the linear transform layer in the *Update* step, with the aim to systematically analyse the impact of the *Update* step. We begin in Section 3.1 by introducing the general model structure of MPNNs, the main GNN class we study in this paper, and expander graphs, the tool we use to sparsify linear layers. We then demonstrate how expander graphs are used to sparsify linear layers and how an *Expander* GNN is constructed in Section 3.2. The idea of *Activation-Only* GNNs is discussed in Section 3.3 and a comparison to the SGC model is drawn.

### 3.1 PRELIMINARIES

#### 3.1.1 MESSAGE-PASSING GRAPH NEURAL NETWORK

We define graphs $\mathcal{G} = (\boldsymbol{A}, \boldsymbol{X})$ in terms of their adjacency matrix $\boldsymbol{A} = [0, 1]^{n \times n}$, which contains the information of the graph's node set $\mathbb{V}$, and the node features $\boldsymbol{X} \in \mathbb{R}^{n \times d} = [\boldsymbol{x}_1, \ldots, \boldsymbol{x}_n]^{\mathrm{T}}$ of dimension $d$. Given a graph $\mathcal{G}$, a graph learning task aims at learning meaningful embeddings on the node or graph level that can be used in downstream tasks such as node or graph classification. MPNNs, a prominent paradigm that arose in recent years for performing machine learning tasks on graphs, learn such embeddings by iteratively aggregating information from the neighbourhoods of each node and updating their representations based on this information. Precisely, the learning procedure of MPNNs can be divided into the following phases:

**Initial (optional).** In this phase, the initial node features $\boldsymbol{X}$ are mapped from the feature space to a hidden space by a parameterised neural network $U^{(0)}$, usually a fully-connected linear layer. $\boldsymbol{H}^{(1)} = U^{(0)}(\boldsymbol{X}) = (\boldsymbol{h}_1^{(1)}, \ldots, \boldsymbol{h}_n^{(1)})$, where the hidden representation of node $i$ is denoted as $\boldsymbol{h}_i^{(1)}$, which will be used as the initial point for later iterations.

**Aggregation.** In this phase, MPNNs gather, for each node, information from the node's neighbourhood, denoted $\mathcal{N}(i)$ for node $i$. The gathered pieces of information are called "messages", denoted by $\boldsymbol{m}_i$. Formally, if $f^{(l)}(\cdot)$ denotes the aggregation function at iteration $l$, then

$$\boldsymbol{m}_i^{(l)} = f^{(l)}(\{\boldsymbol{h}_j^{(l)} | j \in \mathcal{N}(i)\}). \tag{1}$$

Due to the isotropic nature of graphs (arbitrary node labelling), this function needs to be permutation invariant. It also has to be differentiable so that the framework will be end-to-end trainable.

**Update.** The nodes then update their hidden representations based on their current representations and the received "messages". Let $U^{(l)}$ denote the update function at iteration $l$. For node $i$, we have

$$\boldsymbol{h}_i^{(l+1)} = U^{(l)}(\boldsymbol{h}_i^{(l)}, \boldsymbol{m}_i^{(l)}). \tag{2}$$

**Readout (optional).** After $L$ aggregation and update iterations, depending on the downstream tasks, the MPNN will either output node representations directly or generate a graph representation via a readout phase, $\boldsymbol{g} = R(\{\boldsymbol{h}_i^{(L)} | i \in \mathbb{V}\})$. Just like the aggregation function, the readout function also needs to be permutation invariant and differentiable.

This paper focuses on the *Update* step. More precisely, we would like to find the answer to the question: *What is the impact of the Update step on the performance of GNNs?*

It is clear that the MPNN framework divides its learning procedure into two parts: the *Aggregation* step that utilises graph structure and the *Update* step, the main source of model parameters while being completely agnostic to graph structure. Thus, understanding the importance of the *Update* step could have a great impact on the design of parsimonious GNNs.

Note that various choices of *Aggregation*, *Update* and *Readout* functions are proposed in the literature. To avoid shifting from the subject of this paper, we work with the simplest and most widely used function choices, such as sum, mean and max aggregators for *Aggregation*, the Multi-Layer Perceptron (MLP) for the *Update* step and summation for the *Readout*. As an example to visualise our model in the subsequent sections we use the following matrix representation of the GCN computations,

$$\boldsymbol{H}^{(L)} = \sigma(\hat{\boldsymbol{A}} \ldots \sigma(\hat{\boldsymbol{A}} \boldsymbol{H}^{(1)} \boldsymbol{W}^{(1)}) \ldots \boldsymbol{W}^{(L)}), \tag{3}$$

where $\sigma$ denotes a nonlinear activation function, $\boldsymbol{W}^{(i)}$ contains the trainable weights of the linear transform in the update step and $\hat{\boldsymbol{A}} = \tilde{\boldsymbol{D}}^{-\frac{1}{2}} \tilde{\boldsymbol{A}} \tilde{\boldsymbol{D}}^{-\frac{1}{2}}$ is the symmetric normalised adjacency matrix with $\tilde{\boldsymbol{A}} = A + I$ denoting the adjacency matrix with added self-loops and $\tilde{\boldsymbol{D}}$ being the corresponding degree matrix.

Now we proceed to define the class of expander graphs, which serves as necessary background knowledge for our chosen sparsification mechanism in the *Update* step.

### 3.1.2 EXPANDER GRAPHS

Expander graphs are a well studied class of graphs. They can be informally defined as being highly connected and sparse, i.e., in expander graphs relatively few edges are present and arranged in such a way that many edges have to be cut to disconnect the graph (the cut definition is only one way in which the "high connectivity" of expanders can be measured) (Hoory et al., 2006; Lubotzky, 2012). Formally, expander graphs can be characterised by the *expansion ratio*, which we will now define.

**Definition 1** (Expander Graph). For $0 < \delta \in \mathbb{R}$, $\mathcal{G}$ is an $\delta$-*expander graph* if for all $\mathbb{S} \subset \mathbb{V}$ such that $|\mathbb{S}| \leq \frac{|\mathbb{V}|}{2}$ we have $\frac{|\partial\mathbb{S}|}{|\mathbb{S}|} \leq \delta$, where $|\mathbb{S}|$ denotes the cardinality of set $\mathbb{S}$ and $\partial\mathbb{S}$ is the boundary set of $\mathbb{S}$, i.e., the set of all vertices, which are connected to a vertex in $\mathbb{S}$ by an edge but are not in $\mathbb{S}$. Then, the *expansion ratio* $h(\mathcal{G})$ is defined to be the minimal $\delta$ such that $\mathcal{G}$ is an $\delta$-expander graph.

Expander graphs have been successfully applied in communication networks where communication comes at a certain cost and is to be used such that messages are spread across the network efficiently (Lubotzky, 2012). We believe that expander graphs have a promising future in the neural network literature. In a neural network each parameter (corresponding to an edge in the neural network architecture) incurs a comuputational cost both when training the model and when obtaining inference. Therefore, the number of parameters in the model is directly linked to the computational effort and thereby, energy, the model requires to run. Designing neural network architectures using the concept of expander graphs leads to neural network architectures, where fewer edges are placed in such a way that the overall computational structure remains highly connected. In Bölcskei et al. (2019) the connectedness of a sparse neural network architecture was linked to the complexity of a given function class which can be approximated by sparse neural networks. Hence, utilising neural networks parameters to optimise the connectedness of the network maximises the expressivity of the neural network. In Kepner & Robinett (2019) and Bourely et al. (2017) the connectedness of the neural network architecture graph was linked – via the path-connectedness and the graph Laplacian eigenvalues – to the performance of neural network architectures. Therefore, for both the expressivity of the neural network and its performance the connectedness is a parameter of interest and expander graphs utilise a given budget of edges, corresponding to parameters in the neural network, to achieve graphs which are highly connected.

### 3.2 SPARSIFYING THE UPDATE STEP: EXPANDER GNN

In order to study the influence of the *Update* step in GNNs, we propose an experimental design, where its linear transform (of the MLPs) is gradually sparsified. By observing the trend of model performance change (on downstream tasks) with respect to the sparsity of the linear transform layer, we measure the impact of the *Update* step.

**Linear Layer as a graph** The fully-connected linear transform layer in MLPs can be represented by a bipartite graph $\mathcal{B}(\mathbb{S}_1, \mathbb{S}_2, \mathbb{E})$, where $\mathbb{S}_1$ and $\mathbb{S}_2$ are two sets of nodes and $\mathbb{E}$ the set of edges that satisfy $\forall u \in \mathbb{S}_1, \forall v \in \mathbb{S}_2, \exists (u, v) \in \mathbb{E}; \quad \forall u, v \in \mathbb{S}_1 (\text{resp.} \mathbb{S}_2), \nexists (u, v) \in \mathbb{E}$. The number of edges, or number of parameters, is $|\mathbb{S}_1||\mathbb{S}_2|$ and the edges can be encoded in matrix form by $\boldsymbol{W} \in \mathbb{R}^{|\mathbb{S}_1| \times |\mathbb{S}_2|}$, the weight matrix in Equation (3), that maps the input node features of dimension $|\mathbb{S}_1|$ to output node features of dimension $|\mathbb{S}_2|$.

### 3.2.1 EXPANDER LINEAR LAYER

Following Prabhu et al. (2018), we choose expander graphs as the sparsifiers for the linear transform layer. When compared to pruning algorithms which sparsify neural network layers by iteratively removing parameters according to certain metric during training, the expander sparsifiers have two advantages:

(1) Good properties of *expander structures* allow consecutive linear layers to be highly connected when only a smaller number of edges is present. The expander design assures that paths exist between consecutive layers, avoiding the risk of *layer-collapse* that is common in many pruning algorithms, where the algorithm prunes all parameters (weights) in one layer and cuts down the flow between input and output (Tanaka et al., 2020).

(2) The expander sparsifier removes parameters at initialisation and keeps the sparsified structures fixed during training, which avoids the expensive computational cost stemming from adapting the neural network architecture during or after training and then retraining the network as is done in the majority of pruning algorithms (Frankle & Carbin, 2019; Han et al., 2015).

Given the bipartite graph corresponding to a linear transform layer $\mathcal{B}(\mathbb{S}_1, \mathbb{S}_2, \mathbb{E})$, we follow the design of Prabhu et al. (2018) to construct the sparsifier by sampling its subgraph of specific expander structure.

**Definition 2.** Suppose $|\mathbb{S}_1| \le |\mathbb{S}_2|$. For each vertex $u \in \mathbb{S}_1$, we uniformly sample $d$ vertices $\{v_i^u\}_{i=1,\dots,d}$ from $\mathbb{S}_2$ to be connected to $u$. Then, the constructed graph $\mathcal{B}'(\mathbb{S}_1, \mathbb{S}_2, \mathbb{E}')$ is a subgraph of $\mathcal{B}$ with edge set $\mathbb{E}' = \{(u, v_i^u) : u \in \mathbb{S}_1, i \in \{1, \dots, d\}\}$. Else if $|\mathbb{S}_1| > |\mathbb{S}_2|$, we define the expander sparsifier with the roles of $\mathbb{S}_1$ and $\mathbb{S}_2$ reversed meaning that we sample nodes from $\mathbb{S}_1$.

**Definition 3** (layer density). We refer to the $density$ of the expander linear layer as the ratio of the number of sampled connections to the number of connections in the original graph. For example, the fully-connected layer has density 1. The sampling scheme in Definition 2 returns an expander linear layer of density $\frac{d}{|\mathbb{S}_2|}$.

When we replace all linear layers in the *Update* steps of a GNN with expander linear layers constructed by the sampling scheme in Definition 2, we get the *Expander* GNN. An illustration can be found in Appendix B. Bipartite expander graphs as defined in Definition 2 are also discussed in the field of so called "lossless expanders" in Hoory et al. (2006, pp. 517-522), where several expander graph concepts are discussed in the context of bipartite graphs.

### 3.2.2 IMPLEMENTATION OF EXPANDER LINEAR LAYER

The most straightforward way of implementing the expander linear layer is to store the weight matrix $\boldsymbol{W}$ as a sparse matrix. However, due to the known issue of inefficiency of hardware acceleration on sparse matrices (Wen et al., 2016), we use masks, similar to those of pruning algorithms, to achieve the sparsification. A mask $\boldsymbol{M} \in \{0, 1\}^{|\mathbb{S}_1| \times |\mathbb{S}_2|}$ is of the same dimension as weight matrix and $M_{u,v} = 1$ if and only if $(u, v) \in \mathbb{E}'$. An entrywise multiplication is then applied to the mask and the weight matrix so that undesired parameters in the weight matrix are removed, i.e., Equation (3) can be rewritten as,

$$\boldsymbol{H}^{(L)} = \sigma(\hat{\boldsymbol{A}} \dots \sigma(\hat{\boldsymbol{A}} \boldsymbol{H}^{(1)} \boldsymbol{M}^{(1)} \odot \boldsymbol{W}^{(1)}) \dots \boldsymbol{M}^{(L)} \odot \boldsymbol{W}^{(L)}), \tag{4}$$

where $\odot$ denotes the Hadamard product. Theoretically, replacing fully connected linear layers by expander linear layers should both save memory cost and speed up computation. However, this practical implementation, which is adapted to current hardware constraints, worsens the computation time slightly by adding new operations. Contrariwise, the inference computation time is significantly improved by the sparsification. This behaviour of the training and inference time is shared with many pruning approaches, where the training time is increased and the significant time saving comes at the inference stage (Frankle & Carbin, 2019). In Section 4.2 we observe this effect in practice.

### 3.3 An Extreme Case: Activation-Only GNN

A natural extension for the *Expander* GNN is to consider the extreme case where linear transformation layer is removed by removing the trainable weight matrix $\boldsymbol{W}$ from the update step. Gama et al. (2020) argue that the non-linearity present in GNNs, in form of the activation functions, has the effect of frequency mixing in the sense that "part of the energy associated with large eigenvalues" is brought "towards low eigenvalues where it can be discriminated by stable graph filters." This theoretical insight that activation functions help capture information stored in the high energy part of graph signals is strong motivation to consider the extreme case, which we refer to as the *Activation-Only* GNN models, in which each message-passing step is immediately followed by a pointwise activation function and the linear transformation of the update step is forgone. Hence, in a *Activation-Only* GNN, Equation (3) will be rewritten as,

$$\boldsymbol{H}^{(L)} = \sigma(\hat{\boldsymbol{A}} \ldots \sigma(\hat{\boldsymbol{A}} \boldsymbol{H}^{(1)})). \tag{5}$$

This proposed simplification is applicable to a wide variety of GNN models, whose extract formulations can be found in Appendix C.3. For comparison we display the model equation of the SGC (Wu et al., 2019),

$$\boldsymbol{H}^{(L)} = \hat{\boldsymbol{A}}^L \boldsymbol{H}^{(1)} \Theta,$$

where $\Theta = \boldsymbol{W}^{(1)} \ldots \boldsymbol{W}^{(L)}$. Here the nonlinear activation functions have been removed and the linear transformations have been collapsed into a single linear transformation layer. Interestingly, we observe that the repeated application of the symmetric matrix $\hat{\boldsymbol{A}}$ to the input data $\boldsymbol{X}$ is equivalent to an unnormalised version of the power method approximating the eigenvector corresponding to the largest eigenvalue of $\hat{\boldsymbol{A}}$. Hence, if sufficiently many layers $L$ are used then inference is drawn in the SGC model simply on the basis of the first eigenvector of $\hat{\boldsymbol{A}}$.

## 4 Experiments and Discussion

In this section, we empirically study the impact of the *Update* step on model performance. Specifically, in Section 4.1 we provide an overview of the experimentation setup and the vanilla GNNs we compare against. Then, in Sections 4.2, 4.3 and 4.4, we observe the performance of the proposed benchmark models on the tasks of graph classification, graph regression and node classification, respectively. The full set of results can be found in Appendix A.1-A.3. In Section 4.5, we compare the performance of *Expander* GNNs and vanilla GNNs when they have equally many parameters.

### 4.1 General Settings and Baselines

We experiment on eleven datasets from areas such as chemistry, social networks, computer vision and academic citation, for three major graph learning tasks. Details of the used datasets can be found in Appendix C.1. Throughout this section we refer to the standard, already published, architectures as "vanilla" architectures. We compare the performance of the vanilla GNN models, the *Expander* GNN models with different densities ($10\%, 50\%, 90\%$), the *Activation-Only* GNN models with different activation functions (ReLU, PReLU, Tanh), as well as the SGC for the GCN models.

To ensure that our inference is not specific to a certain GNN architecture only, we evaluate the performance across 4 representative GNN models of the literature state-of-the-art. The considered models are the Graph Convolutional Network (GCN, Kipf & Welling (2017)), the Graph Isomorphism Network (GIN, Xu et al. (2019)), the GraphSage Network (Hamilton et al., 2017), and the Principle Neighborhood Aggeragation (PNA, Corso et al. (2020)), along with a MLP baseline that only takes the node features into account while ignoring the graph structure. The *Activation-Only* model class is defined in the context of a GNN architecture and cannot be sensibly extrapolated to the MLP. Therefore, we consider only the vanilla and *Expander* variants for the MLP benchmark.

Since we aim to observe the performance of our benchmark models independent of the GNN choice we use the model hyperparameters found to yield a fair comparison of GNN models in Dwivedi et al. (2020). Other experiment details, such as the choice of loss functions for different tasks, dataset splits as well as the extact message-passing formulation of the models we studied and their variants can be found in Appendix C.

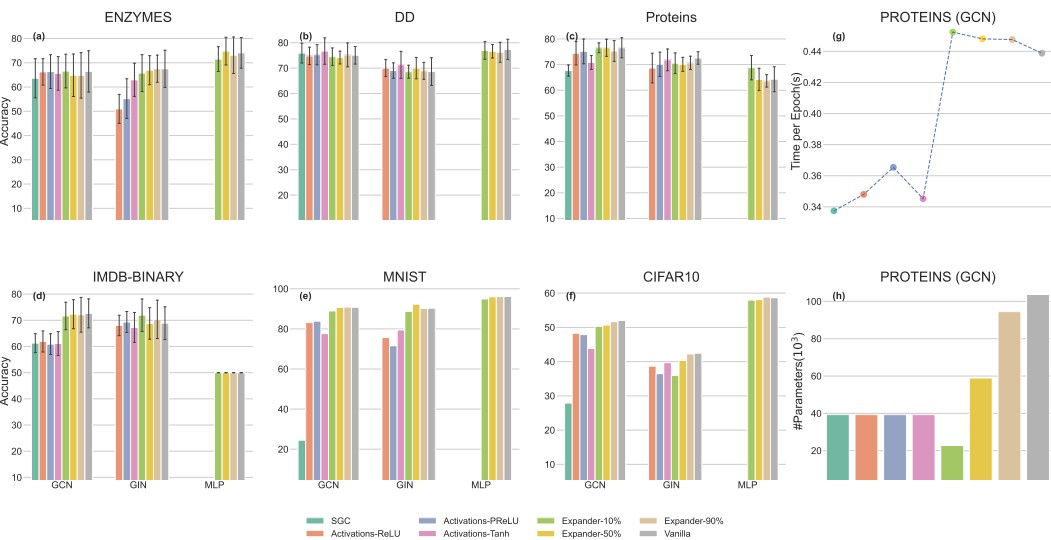

Figure 1: (a,b,c,e): Accuracy of different model types for GCN, GIN and MLP on ENZYMES/DD/PROTEINS/IMDB-BINARY; (f,g): Accuracy of different model types for GCN, GIN and MLP on MNIST/CIFAR10; (d): Training time (per epoch) of different model type for GCN on PROTEINS; (h): Number of parameters required for GCN on PROTEINS.

## 4.2 GRAPH CLASSIFICATION

Figure 1(a), (b), (c) and (e) show the experiment results of the GCN, GIN and MLP models and their *Expander* and *Activation-Only* variants on the ENZYMES, DD, PROTEINS and IMDB-BINARY datasets for graph classification. The evaluation metric is classification accuracy, where the average accuracy, obtained from a 10-folder cross validation, is used.

One direct observation from Figure 1(a), (b), (c) and (e) is that the *Expander* GNN models, even at 10% density, perform on par with the vanilla models. Surprisingly, the same is true for the *Activation-Only* model on the ENZYMES, DD and PROTEINS datasets. IMDB-BINARY is our only graph classification dataset where the node attributes are initialised to all be equal. This uninformative initialisation, surprisingly, seems to lead to an increased performance if the linear update step is present, visible in the performance gap of the *Activation-Only* models and the *Expander* GCN models. While for GIN model we still observe the *Activation-Only* model and to match the performance of the *Expander* GIN. Different activation functions often do not cause significantly different results. However, in a few cases The influence of different activation functions is non-negligible only in a few cases, e.g., in Figure 1(c) the PReLU activation model outperforms the Tanh activation. The SGC performs either on par or worse than the *Activation-Only* model.

It is a known fact that the simple MLP can achieve better performance than GNNs (Luzhnica et al., 2019) on the several of the TU datasets. Dwivedi et al. (2020) have shown that more complex GNN models can outperform the MLP on the aforementioned datasets. However, this known shortcoming has no impact on our conclusion, where we compare model performance within a GNN model class rather than between GNN model classes.

In Figure 1(d) and (h) we observe the *Activation-Only* model to be comparable to the SGC in computation time and in the scale of model parameters. Both models are significantly more efficient than vanilla and *Expander* models. As stated in Section 3.2.2, the training time efficiency of the *Expander* GNN models is slightly less than that of the vanilla models as is expected in the training of a sparsified architecture.

Figure 1(f) and (g) show the graph classification results for the MNIST and CIFAR10 datasets. Interestingly, the GCN *Activation-Only* model outperforms the SGC by a larger margin than we observed on the TU datasets. It seems that especially for these computer vision datasets the presence of activation functions in the GCN architecture has a large positive impact on model performance in the graph classification task.

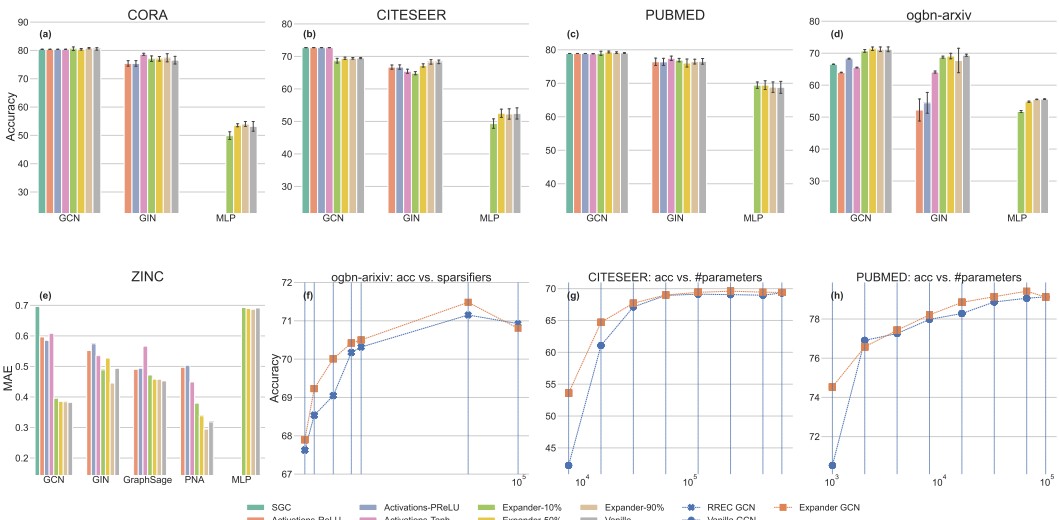

Figure 2: (a,b,c,d): Accuracy of different model types for GCN/GIN/MLP on CORA/CITESEER/PUBMED/ogbn-arxiv datasets; (e): Mean Absolute Error of different model types for GCN/GIN/GraphSage/PNA/MLP on ZINC dataset; (f,g,h): Accuracy vs. number of parameters plot on a logarithmic x-axis on (f) ogbn-arxiv for GCN models with different sparsifiers and (g,h) on CITESEER/PUBMED for vanilla and Expander GCN with same parameter budget.

In summary, almost for all graph classification datasets we find the linear transformation in the update step of the GCN and GIN to make little to no positive contribution to the model performance. The activation function in the update step however turns out to be of great importance on the vision datasets, where as for the TU collection it seems to be of little consequence. Overall, the *Activation-Only* benchmark outperforms the SGC on all observed datasets.

## 4.3 GRAPH REGRESSION

In Figure 2(a) the Mean Absolute Error (MAE) of our studied and proposed models on the ZINC dataset for graph regression is displayed. Similar to the graph classification task, the *Expander* GCN and GraphSage models are on the same level with vanilla models, regardless of their densities. The performance of the *Expander* GIN and PNA models exhibits greater variance accross the different densities, especially in the case of the PNA models the performance is increasing as the network gets denser indicating that the density of the *Update* step does positively contribute to the model performance of the PNA for the task of graph regression on the ZINC dataset. The *Activation-Only* models perform worse than their *Expander* counterparts on this task, again confirming the insight from the results of the *Expander* GNNs that the linear transform in the update step does improve performance in this graph regression task. Again we see that *Activation-Only* GCNs outperform the SGC benchmark in this set of experiments.

Hence, for the task of graph regression we observe that both the linear transformation and non-linear activation function in the Update step have a positive impact on model performance. We might have been able to expect that the addition of the transformation performed in the Update step is of greater impact in a regression task, which is evaluated on a continuous scale, than in a classification task, where only a discrete label needs to be inferred.

## 4.4 NODE CLASSIFICATION

Results from the node classification experiments on four citation graphs (CORA, CITESEER, PUBMED and ogbn-arxiv) can be found in Figure 2(a), (b), (c) and (d), respectively. For medium-sized datasets such as CORA, CITESEER and PUBMED, we have the same observation with the graph classification and graph regression tasks discussed in Sections 4.2 and 4.3, the *Expander* models, regardless of their sparsity, are performing on par with the vanilla ones. Only on the CITESEER

dataset, we observe that the GIN *Expander* model with 10% density shows a small but non-negligible drop in performance compared to the vanilla model; while the 50% and 90% dense *Expander* models remain comparable to the vanilla one. The *Activation-Only* models also perform as well as or even better than (on CITESEER) the vanilla model. The performance of the GCN *Activation-Only* model and SGC are equally good across all three datasets.

These conclusions remain true for large-scale datasets like ogbn-arxiv with 169,343 nodes and 1,166,243 edges. The *ExpanderGCNs* are on par with the vanilla GCN while the *Activation-Only* model and SGC perform slightly worse. However, the training time of *Activation-Only* model and SGC is five times faster than that of the *Expander* and vanilla models. The PReLU *Activation-Only* model performs better than the SGC, while the other two *Activation-Only* models do worse.

We observe that in the node classification task both the linear transformation and the non-linear activation function offer no benefit for the medium scale datasets. For the large-scale dataset we find that the linear transformation can be sparsified heavily without a loss in performance, but deleting it entirely does worsen model performance.

## 4.5 Sparsification vs. Shrinkage

In Figure 2 (f) we compare the results from an *ExpanderGCN* to the those achieved by a GCN model, where the Expander Linear Layer, presented in Definition 2, is replaced by a deterministic sparsificsftion construction from Bourely et al. (2017) the "Regular Rotating Edge Construction" (RREC). Bourely et al. (2017) observe that sparsifiers obtained from the RREC sampler have a significantly lower algrbraic connectivity than the Expander Linear Layer sampler which we chose to utilise in the *Expander* GNNs. We observe that *Expander* GCNs outperform the RREC sampled GCN for almost all parameter budgets. Therefore, we confirm that the Expander Linear Layer is an appropriate choice for the *Expander* GNN model class.

The experiments in Sections 4.2, 4.3 and 4.4 show that the linear transform layer in the Update step of GNN can be sparsified to an arbitrary level without loss of performance. From this a natural question then arises: Will a shrunk model, i.e., a model with a smaller hidden dimension used in the Update step, matching the number of parameters of the sparsified *Expander* GNN, perform on par with its *Expander* GNN counterpart?

To study this question we compare the performance of vanilla GCNs to *Expander* GCNs with equally many parameters, but doubling the size of the hidden dimension of the vanilla GCN. Figure 2 (g) and (h) show the experiment results on two citation datasets. We observe that for most parameter values the *Expander* GCN outperforms the vanilla GCN. This phenomenon becomes more evident when the number of parameter is small. In conclusion, it seems to always be beneficial to choose a sparsified large model rather than a compact model with equally many parameters.

## 5 Conclusion

With extensive experiments across different GNN models and graph learning tasks, we are able to confirm that the *Update* step can be sparsified heavily without a significant performance cost. In fact for seven of the eleven tested datasets across a variety of tasks we found that the linear transform can be removed entirely without a loss in performance, i.e., the *Activation-Only* models performed on par with their vanilla counterparts. The *Activation-Only* GCN model consistently outperformed the SGC model and especially in the computer vision datasets we witnessed that the activation functions seem to be crucial for good model performance accounting for an accuracy difference of up to 59%. These findings partially support the hypothesis by Wu et al. (2019) that the update step can be simplified significantly without a loss in performance. Contrary to Wu et al. (2019) we find that the nonlinear activation functions result in a significant accuracy boost and the linear transformation in the update step can be removed or heavily sparsified.

The *Activation-Only* GNN is an effective and simple benchmark model framework for any message passing neural network. It enables practitioners to test whether they can cut the large amount of model parameters used in the linear transform of the update steps. If the linear transform does contribute positively to the model's performance then the *Expander* GNNs provide a model class of tunable sparsity which allows efficient parameter usage.

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

# APPENDIX

# A COMPLETE EXPERIMENT RESULTS

## A.1 GRAPH CLASSIFICATION

Table 1: Full results of GCN/GIN/MLP on Four TU Datasets

| | | | ENZYMES | | | DD | | |
|---|---|---|---|---|---|---|---|---|
| | | | ACC | Time per Epoch(s) | #Parameters | ACC | Time per Epoch(s) | #Parameters |
| **Simple** | GCN | — | 63.67 ± 8.06 | 0.2004 | 37853 | 75.90 ± 3.93 | 1.3841 | 43643 |
| **Activations** | GCN | lelu | 66.67 ± 6.24 | 0.2073 | | 74.79 ± 3.46 | 1.3979 | |
| | | prelu | 66.67 ± 6.71 | 0.2486 | | 75.39 ± 3.98 | 1.4643 | |
| | | relu | 66.33 ± 5.31 | 0.2132 | 37853 | 74.88 ± 3.41 | 1.3915 | 43643 |
| | | selu | 66.17 ± 6.99 | 0.206 | | 75.64 ± 4.54 | 1.3948 | |
| | | softshrink | 66.17 ± 7.11 | 0.2031 | | 58.66 ± 0.30 | 1.4099 | |
| | | tanh | 65.67 ± 7.04 | 0.2154 | | 76.57 ± 5.20 | 1.391 | |
| | GIN | lelu | 52.17 ± 5.78 | 0.3219 | | 70.12 ± 4.30 | 1.6544 | |
| | | prelu | 55.17 ± 7.94 | 0.3531 | | 69.19 ± 3.02 | 1.7253 | |
| | | relu | 51.00 ± 5.88 | 0.3122 | 5420 | 70.03 ± 3.27 | 1.6424 | 10610 |
| | | selu | 53.83 ± 7.34 | 0.3201 | | 72.49 ± 4.30 | 1.6772 | |
| | | softshrink | 54.83 ± 7.97 | 0.3165 | | 71.22 ± 3.39 | 1.6558 | |
| | | tanh | 62.83 ± 7.15 | 0.3073 | | 71.32 ± 5.29 | 1.6443 | |
| **Expander** | GCN | 10% | 66.33 ± 6.78 | 0.2863 | 22775 | 74.53 ± 3.50 | 1.5731 | 21064 |
| | | 50% | 64.83 ± 8.64 | 0.285 | 58293 | 74.28 ± 2.52 | 1.5856 | 56960 |
| | | 90% | 64.83 ± 9.44 | 0.2917 | 93209 | 75.13 ± 4.69 | 1.5763 | 92215 |
| | GIN | 10% | 65.83 ± 7.75 | 0.3798 | 8918 | 68.59 ± 2.70 | 1.6605 | 6730 |
| | | 50% | 67.00 ± 6.05 | 0.3822 | 29070 | 70.03 ± 4.20 | 1.6427 | 28789 |
| | | 90% | 67.50 ± 5.74 | 0.3729 | 49222 | 68.93 ± 3.26 | 1.637 | 50335 |
| | MLP | 10% | 71.50 ± 5.13 | 0.1653 | 26001 | 77.00 ± 3.39 | 1.0672 | 23858 |
| | | 50% | 74.67 ± 5.72 | 0.1783 | 59661 | 76.66 ± 2.69 | 1.07 | 58020 |
| | | 90% | 73.17 ± 7.65 | 0.1772 | 92811 | 76.24 ± 3.80 | 1.0621 | 91631 |
| **Vanilla** | GCN | — | 66.50 ± 8.71 | 0.2557 | 102239 | 75.13 ± 3.44 | 1.5782 | 101189 |
| | GIN | — | 67.67 ± 7.68 | 0.3692 | 54260 | 68.76 ± 5.55 | 1.6572 | 55978 |
| | MLP | — | 74.17 ± 6.34 | 0.1646 | 101481 | 77.43 ± 3.98 | 1.0523 | 100447 |
| | | | PROTEINS | | | IMDB-BINARY | | |
| | | | ACC | Time per Epoch(s) | #Parameters | ACC | Time per Epoch(s) | #Parameters |
| **Simple** | GCN | — | 67.65 ± 2.21 | 0.3374 | 39311 | 61.30 ± 3.61 | 0.3297 | 31499 |
| **Activations** | GCN | lelu | 75.20 ± 5.19 | 0.3464 | | 61.30 ± 4.05 | 0.3418 | |
| | | prelu | 75.29 ± 4.85 | 0.3654 | | 61.00 ± 3.92 | 0.388 | |
| | | relu | 74.48 ± 4.61 | 0.348 | 39311 | 61.90 ± 4.01 | 0.3497 | 31499 |
| | | selu | 75.92 ± 2.88 | 0.3469 | | 62.70 ± 3.32 | 0.3479 | |
| | | softshrink | 70.53 ± 3.27 | 0.3462 | | 50.00 ± 0.00 | 0.344 | |
| | | tanh | 70.89 ± 2.70 | 0.3452 | | 61.20 ± 4.56 | 0.3522 | |
| | GIN | lelu | 67.74 ± 4.82 | 0.4978 | | 68.10 ± 4.50 | 0.5292 | |
| | | prelu | 70.34 ± 4.78 | 0.5144 | | 69.50 ± 3.88 | 0.577 | |
| | | relu | 68.82 ± 5.97 | 0.4897 | 4410 | 67.90 ± 3.96 | 0.5262 | 1282 |
| | | selu | 72.40 ± 5.03 | 0.4989 | | 69.00 ± 6.03 | 0.5345 | |
| | | softshrink | 69.45 ± 3.66 | 0.4929 | | 69.70 ± 4.31 | 0.5271 | |
| | | tanh | 71.96 ± 4.26 | 0.494 | | 67.30 ± 5.62 | 0.5233 | |
| **Expander** | GCN | 10% | 76.55 ± 1.90 | 0.4524 | 22781 | 71.60 ± 5.50 | 0.4673 | 19920 |
| | | 50% | 76.36 ± 3.43 | 0.448 | 58948 | 72.40 ± 5.70 | 0.4779 | 50888 |
| | | 90% | 75.38 ± 4.01 | 0.4476 | 94502 | 72.30 ± 6.65 | 0.4835 | 81303 |
| | GIN | 10% | 70.53 ± 3.96 | 0.5481 | 6819 | 72.00 ± 6.16 | 0.6368 | 5850 |
| | | 50% | 70.08 ± 2.69 | 0.5483 | 27455 | 68.80 ± 5.90 | 0.6268 | 24125 |
| | | 90% | 70.71 ± 2.55 | 0.5455 | 48091 | 70.30 ± 7.39 | 0.6127 | 41975 |
| | MLP | 10% | 69.00 ± 4.99 | 0.2856 | 25453 | 50.00 ± 0.00 | 0.287 | 22538 |
| | | 50% | 64.15 ± 4.32 | 0.2852 | 58928 | 50.00 ± 0.00 | 0.2896 | 51244 |
| | | 90% | 63.61 ± 2.40 | 0.2831 | 91888 | 50.00 ± 0.00 | 0.2967 | 79487 |
| **Vanilla** | GCN | — | 76.73 ± 3.85 | 0.4388 | 103697 | 72.70 ± 5.68 | 0.4433 | 89045 |
| | GIN | — | 72.51 ± 2.39 | 0.5309 | 53250 | 69.00 ± 6.23 | 0.6135 | 46650 |
| | MLP | — | 64.33 ± 4.93 | 0.2769 | 100643 | 50.00 ± 0.00 | 0.2725 | 86895 |

Table 2: Full results of GCN/GIN/MLP on Computer Vision datasets (MNIST/CIFAR10)

| | | | MNIST | | | CIFAR10 | | |
|---|---|---|---|---|---|---|---|---|
| | | | ACC | Time per Epoch(s) | #Parameters | ACC | Time per Epoch(s) | #Parameters |
| **Simple** | GCN | — | 24.48 | 100.098 | 35811 | 27.90 | 143.721 | 36103 |
| **Activations** | GCN | lelu | 83.52 | 101.554 | 14349 | 48.31 | 39.6657 | 14641 |
| | | prelu | 83.84 | 103.11 | | 47.90 | 41.3494 | |
| | | relu | 83.16 | 102.443 | | 48.27 | 40.2591 | |
| | | tanh | 77.67 | 102.975 | | 43.89 | 40.6699 | |
| | GIN | lelu | 75.68 | 119.803 | 5990 | 37.90 | 44.6216 | 6210 |
| | | prelu | 71.60 | 115.862 | | 36.48 | 46.0211 | |
| | | relu | 75.73 | 119.081 | | 38.67 | 45.0776 | |
| | | tanh | 79.49 | 119.729 | | 39.71 | 45.0382 | |
| **Expander** | GCN | 10% | 89.00 | 124.411 | 22713 | 50.27 | 45.0057 | 22741 |
| | | 50% | 90.75 | 124.075 | 57346 | 50.69 | 45.1236 | 57492 |
| | | 90% | 90.87 | 124.166 | 91392 | 51.68 | 44.769 | 91654 |
| | GIN | 10% | 88.73 | 120.735 | 10973 | 35.93 | 44.4764 | 10995 |
| | | 50% | 92.31 | 119.554 | 30465 | 40.35 | 44.537 | 30575 |
| | | 90% | 90.24 | 116.378 | 49957 | 42.25 | 44.401 | 50155 |
| | MLP | 10% | 94.97 | 67.2461 | 26980 | 57.96 | 31.6988 | 27012 |
| | | 50% | 96.04 | 66.9236 | 61456 | 58.12 | 31.6178 | 61624 |
| | | 90% | 96.17 | 66.7391 | 95425 | 58.85 | 31.6307 | 95727 |
| **Vanilla** | GCN | — | 90.77 | 124.091 | 100197 | 52.04 | 44.337 | 100489 |
| | GIN | — | 90.33 | 120.191 | 54830 | 42.46 | 45.2386 | 55050 |
| | MLP | — | 96.17 | 66.125 | 104044 | 58.66 | 31.5435 | 104380 |

## A.2 GRAPH REGRESSION

Table 3: Full results of GCN/GIN/GraphSage/PNA/MLP on Molecule dataset (ZINC)

| | | | ZINC | | |
| --- | --- | --- | --- | --- | --- |
| | | | MAE | Time per Epoch(s) | #Parameters |
| **Simple** | GCN | — | 0.6963 | 1.9847 | 34347 |
| **Activations** | GCN | lelu | 0.5947 | 1.8624 | 13177 |
| | | prelu | 0.5855 | 1.9273 | |
| | | relu | 0.5967 | 1.8407 | |
| | | selu | 0.6128 | 1.8581 | |
| | | softshrink | 0.6549 | 1.8574 | |
| | | tanh | 0.6086 | 1.8639 | |
| | GIN | lelu | 0.5368 | 2.8046 | 555 |
| | | prelu | 0.5743 | 2.8826 | |
| | | relu | 0.5524 | 2.8009 | |
| | | selu | 0.5424 | 2.8144 | |
| | | softshrink | 0.5221 | 2.821 | |
| | | tanh | 0.5354 | 2.8263 | |
| | GraphSage | lelu | 0.4937 | 4.1406 | 5130 |
| | | prelu | 0.494 | 4.3778 | |
| | | relu | 0.4907 | 4.1238 | |
| | | selu | 0.5365 | 4.1293 | |
| | | softshrink | 1.5508 | 4.1428 | |
| | | tanh | 0.5665 | 4.121 | |
| | PNA | lelu | 0.5181 | 27.9922 | 3515 |
| | | prelu | 0.5038 | 28.1269 | |
| | | relu | 0.4972 | 28.0074 | |
| | | selu | 0.5285 | 27.9645 | |
| | | softshrink | 0.5374 | 28.0993 | |
| | | tanh | 0.4493 | 27.9598 | |
| **Expander** | GCN | 10% | 0.3958 | 2.5833 | 21877 |
| | | 50% | 0.3856 | 2.5793 | 55517 |
| | | 90% | 0.3845 | 2.5578 | 89157 |
| | GIN | 10% | 0.4888 | 3.0905 | 5835 |
| | | 50% | 0.5274 | 3.1125 | 25195 |
| | | 90% | 0.4456 | 3.0852 | 44555 |
| | GraphSage | 10% | 0.4721 | 4.461 | 11970 |
| | | 50% | 0.4584 | 4.4495 | 37890 |
| | | 90% | 0.4579 | 4.4581 | 63810 |
| | MLP | 10% | 0.6931 | 2.1455 | 23775 |
| | | 50% | 0.6898 | 2.1426 | 59775 |
| | | 90% | 0.6873 | 2.1542 | 95775 |
| | PNA | 10% | 0.3798 | 32.0922 | 23735 |
| | | 50% | 0.3384 | 32.148 | 102495 |
| | | 90% | 0.2946 | 32.0666 | 181255 |
| **Vanilla** | GCN | — | 0.3823 | 2.5538 | 97857 |
| | GIN | — | 0.4939 | 3.0569 | 49395 |
| | GraphSage | — | 0.4526 | 4.4149 | 70290 |
| | MLP | — | 0.6916 | 2.1085 | 104775 |
| | PNA | — | 0.3184 | 31.6077 | 201205 |

## A.3 NODE CLASSIFICATION

Table 4: Full results of GCN/GIN/MLP on Three Citations dataset

| | | | CORA | | | CITESEER | | |
|---|---|---|---|---|---|---|---|---|
| | | | ACC | Time per Epoch(s) | #Parameters | ACC | Time per Epoch(s) | #Parameters |
| **Simple** | GCN | — | 80.40 ± 0.00 | 0.0088 | 10038 | 72.70 ± 0.00 | 0.0116 | 22224 |
| **Activations** | GCN | prelu | 80.40 ± 0.00 | 0.0112 | 10038 | 72.70 ± 0.00 | 0.0159 | 22224 |
| | | relu | 80.40 ± 0.00 | 0.0088 | 10038 | 72.70 ± 0.00 | 0.0124 | 22224 |
| | | tanh | 80.40 ± 0.00 | 0.0086 | 10038 | 72.70 ± 0.00 | 0.0125 | 22224 |
| | GIN | prelu | 75.42 ± 0.97 | 0.0124 | 30114 | 66.70 ± 0.65 | 0.0211 | 66672 |
| | | relu | 75.42 ± 0.97 | 0.0117 | 30114 | 66.70 ± 0.65 | 0.0202 | 66672 |
| | | tanh | 78.65 ± 0.36 | 0.0121 | 30114 | 65.45 ± 0.63 | 0.0202 | 66672 |
| **Expander** | GCN | 10% | 80.59 ± 0.64 | 0.021 | 38663 | 68.68 ± 0.73 | 0.0244 | 96518 |
| | | 50% | 80.42 ± 0.28 | 0.0228 | 185351 | 69.43 ± 0.34 | 0.0278 | 475654 |
| | | 90% | 80.82 ± 0.23 | 0.0213 | 332039 | 69.42 ± 0.26 | 0.0259 | 854790 |
| | GIN | 10% | 77.08 ± 0.96 | 0.0163 | 57156 | 64.83 ± 0.49 | 0.022 | 126940 |
| | | 50% | 77.06 ± 0.81 | 0.017 | 230212 | 67.24 ± 0.49 | 0.0253 | 532444 |
| | | 90% | 77.36 ± 1.40 | 0.0167 | 403012 | 68.31 ± 0.70 | 0.0229 | 937692 |
| | MLP | 10% | 49.91 ± 1.34 | 0.0096 | 38663 | 49.34 ± 1.43 | 0.0143 | 96518 |
| | | 50% | 53.57 ± 0.53 | 0.0111 | 185351 | 52.53 ± 1.25 | 0.0175 | 475654 |
| | | 90% | 54.06 ± 0.88 | 0.0102 | 332039 | 52.34 ± 1.64 | 0.0154 | 854790 |
| **Vanilla** | GCN | — | 80.54 ± 0.44 | 0.0172 | 368903 | 69.50 ± 0.19 | 0.0194 | 949766 |
| | GIN | — | 76.57 ± 1.36 | 0.0126 | 446532 | 68.33 ± 0.56 | 0.0154 | 1039324 |
| | MLP | — | 53.21 ± 1.69 | 0.0061 | 368903 | 52.44 ± 1.71 | 0.0086 | 949766 |

| | | | PUBMED | | | ogbn-arxiv | | |
|---|---|---|---|---|---|---|---|---|
| | | | ACC | Time per Epoch(s) | #Parameters | ACC | Time per Epoch(s) | #Parameters |
| **Simple** | GCN | — | 78.90± 0.00 | 0.0148 | 1503 | 66.53± 0.07 | 0.0501 | 5160 |
| **Activations** | GCN | prelu | 78.90± 0.00 | 0.0201 | 1503 | 68.29± 0.13 | 0.0603 | 5160 |
| | | relu | 78.90± 0.00 | 0.0156 | 1503 | 63.97± 0.09 | 0.0519 | 5160 |
| | | tanh | 78.83± 0.05 | 0.0157 | 1503 | 65.51± 0.12 | 0.0518 | 5160 |
| | GIN | prelu | 76.38± 1.06 | 0.0245 | 4509 | 54.45± 3.20 | 0.0732 | 15480 |
| | | relu | 76.46± 1.12 | 0.0235 | 4509 | 52.21± 3.47 | 0.071 | 15480 |
| | | tanh | 77.46± 0.67 | 0.0236 | 4509 | 64.10± 0.32 | 0.0711 | 15480 |
| **Expander** | GCN | 10% | 78.95± 0.63 | 0.0237 | 13827 | 70.70± 0.42 | 0.2733 | 20392 |
| | | 50% | 79.34± 0.28 | 0.0246 | 65027 | 71.42± 0.55 | 0.2681 | 59944 |
| | | 90% | 79.17± 0.23 | 0.0236 | 116227 | 71.22± 0.71 | 0.2732 | 99112 |
| | GIN | 10% | 76.91± 0.51 | 0.0235 | 22757 | 68.78± 0.31 | 0.2742 | 52768 |
| | | 50% | 76.06± 1.14 | 0.024 | 100325 | 69.11± 0.86 | 0.2742 | 118688 |
| | | 90% | 76.50± 0.73 | 0.0236 | 177637 | 67.54± 3.91 | 0.274 | 183968 |
| | MLP | 10% | 69.43± 0.94 | 0.0142 | 13827 | 51.77± 0.30 | 0.1109 | 20392 |
| | | 50% | 69.34± 1.24 | 0.0151 | 65027 | 54.83± 0.20 | 0.1106 | 59944 |
| | | 90% | 68.78± 1.56 | 0.0145 | 116227 | 55.55± 0.08 | 0.1107 | 99112 |
| **Regular** | GCN | — | 79.04± 0.12 | 0.0215 | 129027 | 71.22± 0.76 | 0.2713 | 109096 |
| | GIN | — | 76.55± 0.84 | 0.022 | 197093 | 69.37± 0.34 | 0.2737 | 200608 |
| | MLP | — | 68.87± 1.78 | 0.0129 | 129027 | 55.63± 0.11 | 0.1098 | 109096 |

### A.4 CONVERGE BEHAVIOR: AN EXAMPLE OF GCN ON PROTEINS DATASET

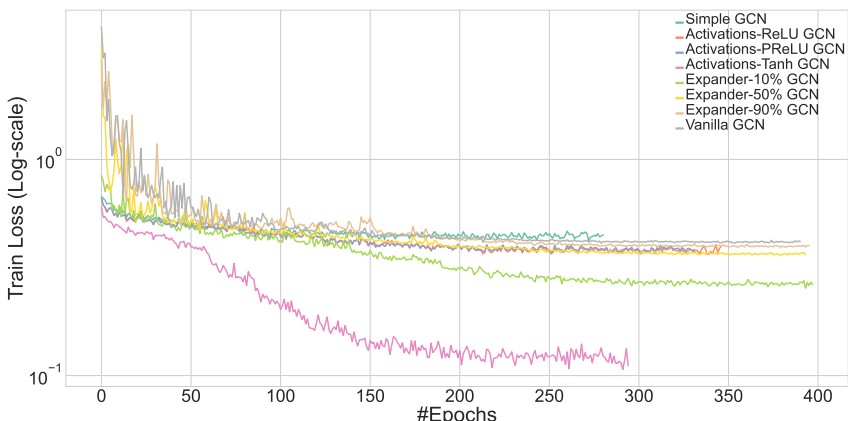

Figure 3: Train loss (cross-entropy) converging behaviour of different model type for GCN on Proteins dataset.

Figure 3 exhibits the loss convergence in training of GCN model class on PROTEINS dataset. We implement a learning rate decay scheme and terminate the training process when learning rate drops to a preset minimum value. Then if a model terminates ealier than its alternatives, it indicates that this model converges faster. As we can see from the figure, *Activation-Only* and SGC both terminate the process with less epochs than vanilla and *Expander* models. Similar to the main paper, we are able to draw the conclusion that *Activation-Only* and SGC do not only have fewer parameters but that their training also converges faster, hence more efficient in time.

## B ILLSTRATION OF EXPANDER MPNNS

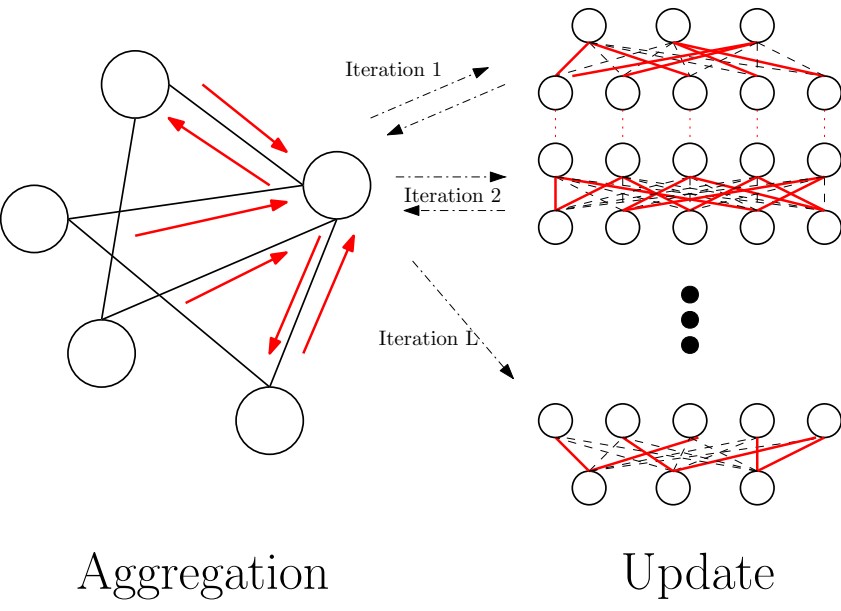

Figure 4: Illustration of *Expander* MPNNs. The left part is the *Aggregation* or graph propagation step and the right part is the *Update* step. The red lines on the left part represents preserved connections in MLPs sampled by expander sparsifier.

# C  EXTRA INFORMATION FOR EXPERIMENTS AND MODELS

## C.1  DATASET DETAILS

Table 5: Properties of all datasets used in experiments.

|  | Dataset | #Graphs | #Nodes (avg.) | #Edges (avg.) | Task |
|---|---|---|---|---|---|
| TU datasets | ENZYMES | 600 | 32.63 | 62.14 | Graph Classification |
|  | DD | 1178 | 284.32 | 715.66 |  |
|  | PROTEINS | 1113 | 39.06 | 72.82 |  |
|  | IMDB-BINARY | 1000 | 19.77 | 193.06 |  |
| Computer Vision | MNIST | 70K | 70.57 | 282.27 |  |
|  | CIFAR10 | 60K | 117.63 | 470.53 |  |
|  | ZINC | 12K | 23.16 | 24.92 | Graph Regression |
| Citations | CORA | 1 | 2708 | 5278 | Node Classification |
|  | CITESEER | 1 | 3327 | 4552 |  |
|  | PUBMED | 1 | 19717 | 44324 |  |
|  | ogbn-arxiv | 1 | 169343 | 1166243 |  |

Differed on the learning tasks, the datasets we use can be classified into three categories. For graph classification, we have four TU datasets (Kersting et al., 2016) which are either chemical or social network graphs, and two Image datasets (MNIST/CIFAR10) that are constructed from original images following the procedure in Knyazev et al. (2019). To perform this conversion they first extract small regions of homogeneous intensity from the images, named "Superpixels" (Dwivedi et al., 2020), and construct a $K$-nearest neighbour graph from these superpixels. The technique we implemented to extract superpixels, the choice of $K$ and distance kernel for constructing a nearest neighbour graph are the same as in Knyazev et al. (2019) and Dwivedi et al. (2020). For graph regression, we consider molecule graphs from the ZINC dataset (Irwin et al., 2012). And for node classification, we use four citation datasets (Sen et al., 2008; Wang et al., 2020; Hu et al., 2020), where the nodes are academic articles linked by citations.

In Table 5, we summarise the statistics of the aforementioned eleven datasets in detail. We display their number of graphs, number of nodes and of edges, as well as the tasks that are performed on them. In the number of nodes and edges column we show the values, or the average of these values if there are multiple graphs.

## C.2  EXPERIMENT SETTINGS

In order to ensure a fair comparison across different GNN models, we follow the recent benchmark proposed in Dwivedi et al. (2020). Specifically, we use their datasets on computer vision (MNIST/CIFAR10) and chemistry(TU datasets/ZINC dataset); we follow the same training procedure, such as train/valid/test dataset splits, choice of optimiser, learning rate decay scheme, etc., as well as the same hyper-parameters, such as initial learning rate, hidden feature dimensions, number of GNN layers, etc. We also implement the same normalisation tricks such as adding batch normalisation after non-linearity of each *Update* step. Their setting files (training procedure/hyperparameters) are made public and can be found in this repository.

For the node classification task on citation datasets, we follow the settings from Wu et al. (2019). Our experiments found that the node classification task on citation graphs of small/medium size can be easily overfit and model performances heavily depend on the choice of hyperparameters. Using the same parameters with Wu et al. (2019), such as learning rate, number of training epochs and number of GNN layers, helps us achieve similar results with the paper on the same model, which allows a fair comparison between the proposed *Activation-Only* models and the SGC.

## C.3  MODEL DETAILS: GNNS UNDER THE MPNN FRAMEWORK

In Section 3, we present the message-passing GNN framework and explained the ideas which drive the *Expander* and *Activation-Only* models that we proposed. These ideas are illustrated by an example of GCN. In this section, we will recall the MPNN formulation. Normalisation layers such as batch normalisation are omitted in the formulation for the purpose of clarity. For each of the models, we

present their vanilla version, *Expander* version and *Activation-Only* version. As stated in Section 3, we ignore the various variants of vanilla models and only work with variants of simple forms.

### C.3.1 GRAPH CONVOLUTIONAL NETWORK

**Vanilla GCN and *Expander* GCN**   We have shown the matrix form of *Aggregation* and *Update* iterations in GCN in the main paper. Here we are going to revisit the message-passing procedure in GCN from the angle of a single node. At iteration $l$,

$$m_i^{(l)} = \frac{1}{\sqrt{\deg_i}} \sum_{j \in \mathcal{N}(i)} h_j^{(l)} \frac{1}{\sqrt{\deg_j}}, \tag{6}$$

$$h_i^{(l+1)} = \sigma(m_i^{(l)} M^{(l)} \odot W^{(l)}), \tag{7}$$

where $W^{(l)}$ is the weight matrix of $l^{\text{th}}$ linear transform layer and $M^{(l)}$ is its corresponding mask. For vanilla GCNs, all entry of $M^{(l)}$ is 1 while for *Expander* GNN $M^{(l)}$ is a sparse matrix. Equation (6) is the *Aggregation* step, which constructs "messgae" equivalent to a weighted average of neighbours' embeddings, then Equation (7) *Updates* node $i$'s representation based on constructed "message".

***Activation-Only* GCN**   The *Activation-Only* model can then simply be written as,

$$h_i^{(l+1)} = \sigma \left( \frac{1}{\sqrt{\deg_i}} \sum_{j \in \mathcal{N}(i)} h_j^{(l)} \frac{1}{\sqrt{\deg_j}} \right).$$

### C.3.2 GRAPH ISOMORPHISM NETWORK

**Vanilla GIN and *Expander* GIN**   The message-passing procedure of GIN is very similar to GCN, except that at the *Aggregation* step, it adds explicitly a learnable ratio of the central node's own representation, defined as,

$$m_i^{(l)} = (1 + \epsilon) h_i^{(l)} + \sum_{j \in \mathcal{N}(i)} h_j^{(l)}. \tag{8}$$

Its *Update* step is the same with Equation (7).

***Activation-Only* GIN**   Similar to GCN, the *Activation-Only* model can then be written as,

$$h_i^{(l+1)} = \sigma \left( (1 + \epsilon) h_i^{(l)} + \sum_{j \in \mathcal{N}(i)} h_j^{(l)} \right).$$

### C.3.3 GRAPHSAGE NETWORK

**Vanilla GraphSAGE and *Expander* GraphSAGE**   GraphSage also incorporates explicitly the central node's representation in the *Aggregation* step by concatenation. Their message-passing procedure is,

$$m_i^{(l)} = h_i^{(l)} \| \text{MAX}_{j \in \mathcal{N}(i)} \sigma(h_j^{(l)} M_1^{(l)} \odot W_1^{(l)}), \tag{9}$$

$$h_i^{(l+1)} = \frac{\hat{h}_i^{(l+1)}}{\|\hat{h}_i^{(l+1)}\|_2}, \quad \hat{h}_i^{(l+1)} = \sigma(m_i^{(l)} M_2^{(l)} \odot W_2^{(l)}), \tag{10}$$

where $\|$ denotes concatenation and $\text{MAX}$ function takes maximum along each feature dimension.

***Activation-Only* GraphSAGE**   In *Activation-Only* models for GraphSage, we need to consider the issue of dimension incoherence after removing linear transforms. Since a simple removal of linear transforms will result in an exponential growth in the dimension of hidden representation. One

convenient solution is to replace concatenation with a summation so that the dimension of either $\boldsymbol{m}$ or $\boldsymbol{h}$ remains unchanged after the iteration. The message-passing steps then become,

$$\boldsymbol{m}_i^{(l)} = \boldsymbol{h}_i^{(l)} + \mathrm{MAX}_{j \in \mathcal{N}(i)} \sigma(\boldsymbol{h}_j^{(l)}), \tag{11}$$

$$\boldsymbol{h}_i^{(l+1)} = \frac{\hat{\boldsymbol{h}}_i^{(l+1)}}{\|\hat{\boldsymbol{h}}_i^{(l+1)}\|_2}, \quad \hat{\boldsymbol{h}}_i^{(l+1)} = \sigma(\boldsymbol{m}_i^{(l)}). \tag{12}$$

A more complex way is to separate the propagation and update of "message" $\boldsymbol{m}$ and hidden representation $\boldsymbol{h}$. More precisely,

$$\boldsymbol{m}_i^{(l)} = \mathrm{MAX}_{j \in \mathcal{N}(i)} \sigma(\boldsymbol{m}_j^{(l-1)}), \quad \boldsymbol{m}_i^{(0)} = \boldsymbol{h}_i^{(1)}, \tag{13}$$

$$\boldsymbol{h}_i^{(l+1)} = \frac{\hat{\boldsymbol{h}}_i^{(l+1)}}{\|\hat{\boldsymbol{h}}_i^{(l+1)}\|_2}, \quad \hat{\boldsymbol{h}}_i^{(l+1)} = \sigma(\boldsymbol{h}_i^{(l)}\|\boldsymbol{m}_i^{(l)}), \tag{14}$$

where the dimension of $\boldsymbol{h}_i$ grows proportionally to the number of iterations. Neither of the methods is out-weighted by the other based on evidence from our empirical experiments. However, from an economic view on developping parsimonious GNN, Equation 11 is our preferred method.

### C.3.4  PRINCIPAL NEIGHBOURHOOD AGGREGATION

**Vanilla PNA and *Expander* PNA**    The PNA model concatenates in its *Aggregation* step the "messages" obtained from different combinations of scalars and aggregators. This step can be written as (Fey & Lenssen, 2019),

$$\boldsymbol{m}_i^{(l)} = \bigoplus_{j \in \mathcal{N}(i)} \sigma(\boldsymbol{h}_j^{(l)} \boldsymbol{M}^{(l)} \odot \boldsymbol{W}^{(l)}), \tag{15}$$

where $\bigoplus$ is defined as the outer product, denoted by $\otimes$, of the arrays of scalars and aggregators, with cardinality $c_1$ and $c_2$, respectively, as follow,

$$\underbrace{\begin{bmatrix} 1 \\ S(\mathbf{D}, \alpha = 1) \\ S(\mathbf{D}, \alpha = -1) \end{bmatrix}}_{\text{scalars}} \otimes \underbrace{\begin{bmatrix} \text{mean} \\ \text{std} \\ \text{max} \\ \text{min} \end{bmatrix}}_{\text{aggregators}}.$$

The *Update* step is the same as Equation (7).

**Activation-Only PNA**    Similar to the *Activation-Only* GraphSage, the *Activation-Only* PNA also suffers from dimension incoherence issue, due to the concatenate operation. Instead of concatenation, we take the average of each representation obtained from one combination of scaler and aggregator in *Activation-Only* models. Then the message-passing procedure becomes,

$$\boldsymbol{h}_i^{(l+1)} = \sigma \left( \frac{1}{c_1 c_2} \sum \mathbf{1}^T \left[ \bigoplus_{j \in \mathcal{N}(i)} \sigma(\boldsymbol{h}_j^{(l)}) \right] \mathbf{1} \right),$$

where $c_1$ denotes the number of scalars and $c_2$ denotes the number of aggregators used in the PNA.

### C.3.5  GRAPH-AGNOSTIC BASELINE: MULTI-LAYER PERCEPTRON

**Vanilla MLP and *Expander* MLP**    The MLP baseline has no *Aggregation* steps. Its *Update* step is simply,

$$\boldsymbol{h}_i^{(l+1)} = \sigma(\boldsymbol{h}_i^{(l)} \boldsymbol{M}^{(l)} \odot \boldsymbol{W}^{(l)}).$$

## C.4 Loss functions for different tasks

After $L$ message-passing iterations, we obtain $\boldsymbol{H}^{(L)} = [\boldsymbol{h}_1^{(L)}, \ldots, \boldsymbol{h}_n^{(L)}]^{\mathrm{T}} \in \mathbb{R}^{n \times d}$ as the final node embedding, where we denote $d$ as its feature dimension. Depending on downstream tasks, we either keep working with $\boldsymbol{H}^{(L)}$ or construct a graph-level representation $\boldsymbol{g}$ from $\boldsymbol{H}^{(L)}$ by,

$$\boldsymbol{g} = \frac{1}{n} \sum_{i \in \mathbb{V}} \boldsymbol{h}_i^{(L)},$$

which is refer to as the *Readout* step in Section 3.

$\boldsymbol{g}$ or $\boldsymbol{H}^{(L)}$ is then fed into a fully-connected network (MLP) to be transformed into the desired form of output for further assessment, i.e., a scalar value as prediction scores for graph regression. We denote this network as $f(\cdot)$, which, in our experiments, is fixed to be a three-layer MLP of the form

$$f(\cdot) = \sigma(\sigma(\cdot, \boldsymbol{W}_1)\boldsymbol{W}_2)\boldsymbol{W}_3,$$

where $\boldsymbol{W}_1 \in \mathbb{R}^{d \times \lceil \frac{d}{2} \rceil}$, $\boldsymbol{W}_2 \in \mathbb{R}^{\lceil \frac{d}{2} \rceil \times \lceil \frac{d}{4} \rceil}$, $\boldsymbol{W}_3 \in \mathbb{R}^{\lceil \frac{d}{4} \rceil \times k}$ with $k$ being the desired output dimension.

The final output, either $f(\boldsymbol{g})$ or $f(\boldsymbol{H}^{(L)})$, is compared to the ground-truth by a task-specific loss function. For graph classification and node classification, we choose cross-entropy loss and for graph regression, we use mean absolute error (or the $L1$ loss).

