# OpenReview forum: "Analysing the Update step in Graph Neural Networks via Sparsification"
_ICLR.cc/2021/Conference — Reject_

### Official Review · AnonReviewer3 · 2020-10-25
**REVIEW FOR "ANALYSING THE UPDATE STEP IN GRAPH NEURAL NETWORKS VIA SPARSIFICATION"**

**Rating:** 4
**Confidence:** 4

**Review:**

This paper proposed to sparsify the update step of MPNN (message passing neural network)-typed graph neural networks. Specifically, the authors borrowed the idea of upsampling from X-net in CNN and update step simplification from SGC. Although with some presentation issues, the general idea is easy to follow. The reported empirical prediction performance of the simplified models on several standard datasets is comparable to that of the baseline models. However, I have some concerns regarding the methodology and the experimental settings, which, at this stage.

**Below listed are the main concerns or confusion regarding the paper.
1. (Section 3.2) what’s the meaning of $|\mathbb{S}_1|$ and $|\mathbb{S}_2|$? By absolute, does it mean the number of nodes in the two sets? Also how is the size of $|\mathbb{S}_1||\mathbb{S}_2|$ defined? Do you mean $|\mathbb{S}_1|*|\mathbb{S}_2|$?
2. (Section 3.2.1) When mentioning the pruning algorithms, it would be more appropriate to add related citations after the concept.
3. (Section 3.2.1) As mentioned in the second point, the expander sparsifiers are claimed to 'avoid the expensive computational cost'. While there are no reputable citations over the statement, one would expect some specific evidence to support it. For example, what is the computational cost gap between the pruning methods and the expender sparsifiers?
4. (Section 3.2.1) As $\mathcal{B}$ is interpreted as a linear transform, it would be clearer to define from which space to which space it transforms the information.
5. (Definition 2) What does it mean by 'without loss of generality'?
6. (Definition 2) What is the size of the adjacency matrix $\mathbf{A}' $? Do you define $\mathbf{A}^{
\prime} \in \mathcal{R}^{d*d}$, $\mathbf{A}^{\prime} \in \mathcal{R}^{|\mathbb{S}_1|*d}$, or some other specific dimensions? Also, are there any constraints on setting the value of $d$?
7. (Section 3.3) It is not clear how Equation 4 is ended up with an 'activation-only' form when the density is 0, i.e., $d=0$. The $d=0$ would result in a zero matrix of $\mathbf{M}$, which cuts off all the signal transformations within the network. As a result, the output $\mathbf{H}^{L}$ should also be a zero matrix. If different update rules are used when $d=0$, it should be clearly stated, rather than 'deriving from the general case of Equation 4'.
8. The figures are difficult to read or understand. First, the labels are ambiguous to me. For example, which 'simple' and 'vanilla' models are referred to exactly? Also, why the number of bars for each of the methods (GCN, GIN, MLP) are different? Why some of the datasets (ENZYMES, DD, etc.) include error bars while others do not? It is recommended for a more straightforward and informative presentation to substitute the figures with tables, which provide readable comparable numeric values.
9. The experimental results, especially displayed in Figure 1, cannot suggest the proposed methods' power. According to Figure 1-(d) and 1-(f), the last 3 Expander methods (except expander-10\%) require more time and parameters to obtain a 'comparable' accuracy. Adopting such a structure is then in question.
10. The results in Figure 1 are surprising, where MLP based methods outperform other models significantly. What is the possible explanation of it?
**It is also found that many sentences are difficult to understand due to the over-complicated expressions, expositional problems, and/or grammatical mistakes. Some examples which need improvement:
1. (line1, Section 3.1.1) "Given a graph…": can be simplified to shorter sentences.
2. (line 4, page 3) Maybe some grammar problems in "The idea of … has found successful application".
3. (line 1, Section 3.1.2) "Expander graphs…can be informally defined as being highly connected and sparse". The meaning of "sparse" is not clear.
4. (line 12, Section 3.1.2) "Training each parameter … incurs some training and inference, computational". This sentence is ambiguous.
5. (line 13, Section 3.1.2) "Using the expander graph concept … reducing thus cost, and it is the aim to utilise each parameter to its greatest efficiency". The meaning of "reducing thus cost" and 'greatest efficiency' is not clear.

Given the problems stated above, the paper needs significant improvement and polishment before it can be considered acceptable.

---

> ### Author Response · Authors · 2020-11-25
> **Response to Reviewer 3: Significant improvements (1/2)**
>
> Thank you very much for your thorough and constructive review. Your review has helped us to make substantial additions and changes to the paper, which, as we hope you will agree, have significantly improved and polished the paper.  We have uploaded an accordingly revised manuscript. We would now like to present a more detailed discussion of the points you raised, following the order in which you presented them.
>
>
> 1. We have added the definition of this notation to its first occurrence in Definition 1. Specifically, we have added “$\lvert \mathbb{S} \rvert$ denotes the cardinality of the set $\mathbb{S}$”. Indeed the term $\lvert \mathbb{S}_1 \rvert \lvert \mathbb{S}_2 \rvert $ refers to the product of  $\lvert \mathbb{S}_1 \rvert$ and $ \lvert \mathbb{S}_2 \rvert.$ We believe that the explicit definition of the cardinality should also clear up this question without further discussion in the paper.
>
>
> 2. Thank you very much for the suggestion of adding pruning citations. We agree that in this way the statements are strengthened significantly. We have added two appropriate references to Section 3.2.1.
>
>
> 3. Pruning algorithms generally train a dense architecture and then determine weights to be removed. Often the removal of weights is followed by retraining the neural network and potentially further steps. In contrast, our Expander GNN architecture initialises a sparse architecture which is trained once. Since both approaches involve training a neural network and the majority of  pruning approaches incorporate further steps in addition to the training, the ExpanderGNNs save computational cost in comparison. We have now included a more explicit discussion of this rationale in Section 3.2.1.
>
>
> 4. In fact $\mathcal{B}(\mathbb{S}_1, \mathbb{S}_2, \mathbb{E})$ denotes the bipartite graph corresponding to a linear transform layer. Therefore, $\mathcal{B}(\mathbb{S}_1, \mathbb{S}_2, \mathbb{E})$ itself does not transform node features from one space to another. However, the corresponding weight matrix encoding the edges in $\mathcal{B}(\mathbb{S}_1, \mathbb{S}_2, \mathbb{E})$ does map input node features of dimension $|\mathbb{S}_1|$ to output node features of dimension $|\mathbb{S}_2|.$ We have improved the text in Section 3.2 and 3.2.1 to clarify this representation.
>
>
> 5. We agree the use of this term is confusing in Defintion 2 and we have therefore reformulated it to present the concept more clearly.
>
>
> 6. Since the linear layer has a total of $\lvert \mathbb{S}_1 \rvert + \lvert \mathbb{S}_2 \rvert $ nodes, the adjacency matrix corresponding to the bipartite graph representing this linear layer is a square matrix with $\lvert \mathbb{S}_1 \rvert + \lvert \mathbb{S}_2 \rvert $ rows and columns. However, we have now reformulated Definition 2 to not rely on the adjacency matrix and instead be defined in terms of the edge set. We hope that this change helps clear up this confusion. In response to your second question, in Definition 2 we state that we sample $d$ vertices from $\mathbb{S}_2$ and as a consequence $d$ is greater than 0 and smaller than $ \lvert \mathbb{S}_2 \rvert.$ In this theoretical discussion we choose not to impose further constraints on $d$ and in practice we observe in Section 4 that ExpanderGNNs with $d=0.1\lvert \mathbb{S}_2 \rvert$ perform on par with the vanilla GNNs in many datasets and tasks.
>
>
> 7. The precise calculations performed in the Activation-Only models are given in Equation (5). We see how introducing the Activation-Only as an ExpanderGNN with density 0 in the update step is misleading and have therefore adapted the introduction in Section 3.3 to remove this mistake.
>
>
> 8. We have replaced the figure label “simple” with the label “SGC” to be more explicit. We have furthermore included a sentence in Section 4.1 clarifying that we refer to the standard, already published, architectures as vanilla models throughout the section. In addition, we have included an explanation that the Activation-Only benchmark is not well defined for a MLP and specific to GNN models. This should clear up your question about the different number of benchmarks available for the different architectures. We have added error bars on to all experiments except the vision datasets, due to the computation cost involved in the repetition of experiments to obtain the error bars. We agree that in tables we are able to more clearly demonstrate the results. Due to the large number of experiments and the page limit we present the tables containing the full results to Appendix A and we utilise figures in the main body of the text to give an overview of the results. We have now included a note in the introduction of Section 4 pointing out this appendix to the reader.

---

> > ### Author Response · Authors · 2020-11-25
> > **Response to Reviewer 3: Significant improvements (2/2)**
> >
> > 9. We agree that, regrettably, the presentation of the results in the originally submitted manuscript did not seem practical. In fact, the increased training time (with respect to the vanilla model) is unavoidable with the current hardware limitations and shared with almost all existing pruning algorithms. The reason why sparsification is such an active research area is that sparsified models enable significant time savings when used for inference, where the multiplication of sparse weight matrices leads to significant time savings. Therefore, from a computation cost standpoint there is a strong motivation to use our proposed benchmark models. Of course the Activation-Only model class achieves better computation times in training and inference and is therefore, clearly a feasible benchmark model. We have adapted the discussion of these results in Section 4.2 to clarify the practical benefits of our models.
> > Regarding the number of parameters, the Expander GNN, as well as the  Activation-Only model require strictly fewer parameters than the vanilla model (grey bar on the right in Figure 1 (h)). We have now updated our discussion to incorporate this additional information  in Sections 3.2.2 and 4.2.
> >
> >
> > 10. This is a good question and we agree that the necessary discussion was missing from our manuscript. For several of the chosen datasets (e.g. ENZYMES and DD) it is a known fact that  the simple MLP can achieve better performance than GNNs [1]. A recent study [2] has shown that more complex GNN models can outperform the MLP on the aforementioned datasets. However, this known shortcoming has no impact on our conclusion. The objective of the experiments is not to compare between different GNN models and find out the highest performing, but rather to compare within each GNN model, its simplified, sparsified and vanilla versions, in order to find out the importance of  each component of the update step. And our finding is unanimous across the majority of the chosen datasets: the linear transform in the Update step of GNNs can be sparsified to an arbitrary level and can even be omitted in many cases without a loss of performance.  We have included this discussion of our datasets in Section 4.2.
> >
> >
> > In the revised manuscript, we have furthermore addressed the five syntactic issues you identified.
> >
> >
> > [1] Enxhell Luzhnica, Ben Day & Pietro Liò, “On Graph Classification Networks, Datasets and Baselines,” In: Workshop on Learning and Reasoning with Graph-Structured Data (ICML),  2019.
> >
> > [2] Vijay Prakash Dwivedi, Chaitanya K. Joshi, Thomas Laurent, Yoshua Bengio & Xavier Bresson, “Benchmarking graph neural networks,” arXiv:2003.00982, 2020.

---

### Official Review · AnonReviewer4 · 2020-10-26
**The paper introduces the use of simplified Update steps in GNNs and shows on par performance with the simplified models. Discussion is lacking and contribution is of concern.**

**Rating:** 5
**Confidence:** 3

**Review:**

#### **Post Rebuttal**
I thank the authors for the elaborated answers and additional experiments. Unfortunately, I believe that although this work introduces some interesting insights it needs a little more work to achieve the ICLR publication bar.
I therefore keep my original score.
---

#### **Summary of Contribution Claims**
The paper questions the necessity of complicated and computationally expensive  dense Update steps in message passing GNNs.
To answer the question two GNN variants with simple Update steps, expander based sparse and activation only, are suggested.
The paper presents a wide empirical evaluation on various datasets, demonstrating the abilities of the suggested models,  showing that in many cases, a simple Update step is good enough. From a practitioner point of view, the activation only model can serve as a simple "test" model to check whether on a specific task, a linear transformation in the Update step is necessary.

**Strengths**
- Clarity of ideas - the ideas behind the design of suggested Update steps are clearly explained and motivated.
- Novel observation - in some cases, even the extreme case of an activation only Update step achieves good results.

**Weaknesses**
- Novelty and Contribution - I have some concerns regarding the novelty and contribution of this paper. As the use of expander graphs to sparsify NN layers has already been used in other NN architectures, I would say that Expander GNN is an extension. Furthermore, since the paper leans towards improving the practical side of GNNs, I would say that the hardware limitations mentioned in the paper are of concern.

- Motivation vs. Evaluation - the paper motivates the design of sparse Update steps by the reduced computational costs. However, in the experiments, only relatively small graph datasets are evaluated on, leaving, in my opinion, the question of the effectivity of this approach in the desired setting open.

- Experiments -  as mentioned above, I feel that the datasets over which the method has been evaluated do not provide a convincing evidence to the effectiveness of the approach. I would like to see performance on some datasets used in the SGC paper.

- Discussion - the analysis and discussion of the results is rather a translation of the presented graphs without providing insights and possible explanations for the observed phenomena. From the results discussion in the paper I take that the success of the simplified Update steps is task dependent however, I would like to have some understanding and characterization as to when or why things work the way they do?

#### **Decision Recommendation**
Although presenting interesting empirical results, and potentially valuable simplifications of GNNs, I feel that this paper can be improved by performing evaluation on more relevant datasets, and acquiring stronger theoretical understanding and explanation of the observed results.
I would suggest to reject the paper.

#### **Other Comments**
The paper title is a bit misleading, I would suggest changing Analyzing to Simplifying, since the paper mainly proposes the method and shows on par performance,  but the analysis is lacking.

---

> ### Author Response · Authors · 2020-11-25
> **Response to Reviewer 4: New dataset and theorectical analysis (1/2)**
>
> Thank you very much for your insightful review. On the basis of your comments have been able to make significant improvements and additions to paper, especially in Section 4. We have uploaded a revised manuscript where these changes have been made. We would now like to give detailed responses to the weaknesses you identified in the order that you presented them.
>
> 1. *“Novelty and Contribution”*
> We completely agree that the sparsification design of the ExpanderGNN relies on the work by Prabhu et al [1]. However, the Activation-Only model is an original contribution made by us in this paper. The main contribution of this paper lies in the comprehensive empirical study and the insight into the practical impact of each component of the Update step.
> In our originally submitted manuscript we, regrettably, failed to mention that the increased training time is a property of almost all existing pruning algorithms. Therefore, the slightly increased training time of our sparsified Expander GNNs is not a weakness specific to the proposed model class, but rather of the currently available hardware. The reason why sparsification is such an active research area is that sparsified models enable significant time savings when used for inference, which outweighs the additional cost at training. We have now updated our discussion of the computation time of our model to reflect this discussion in Sections 3.2.2 and 4.2.
>
>
>
> 2. *“Motivation vs. Evaluation"*
> We agree that the effectivity of our approach is more clearly demonstrated on a large graph dataset. We have therefore added experiments on OGBN-Arxiv, which is a large graph dataset (169,343 nodes, 1,166,243 edges) for node classification tasks from the Open Graph Benchmark project [2]. The results can be found in Section 4.4 Figure 3. They are consistent with previous experiments and strengthen our conclusion. We chose our remaining datasets since they are the most widely used datasets in the GNN literature and thus, are good reference points against which to measure the performance of our proposed model.
>
>
> 3. *“Experiments”*
> In fact, there is already a significant overlap between the datasets used in the SGC paper and our manuscript. We both used the same citation datasets (CORA/CITESEER/PUBMED),  where the SGC was proven to be very effective. Since the publication of the SGC paper, the GNN benchmark datasets have evolved and this is reflected in our choice of datasets. We chose to follow [3] and experiment on datasets corresponding to a variety of learning tasks and data sources to carry out a fair comparison between the SGC and our proposed models.
> On the chosen datasets we are able to observe that in several tasks and datasets the SGC is far outperformed by the Activation-Only benchmarks which matches the SGC’s runtime and number of parameters. We are furthermore able to draw conclusions on the impact of the different components of the Update step on three graph learning tasks. Therefore, we argue that the chosen datasets do enable us to draw convincing conclusions from the experiments.
>
>
> 4. *“Discussion”*
> Thank you very much for pointing out the lack of interpretation and analysis in Section 4 out to us. We have made significant additions to Sections 4.2, 4.3 and 4.4, where we provide our interpretation of the results and how they reflect on the impact of the Update step in practice.
> It seems that on the graph regression task, which is evaluated on a continuous scale, both the non-linear activation function and the linear transform in the Update step have an impact on model performance in practice. In the graph classification task, which is less sensitive, since it is evaluated on a discrete scale, we find that for almost all datasets the linear transformation in the Update step has little to no positive impact on the model performance. The activation function in the Update step however turns out to be of great importance on the vision datasets, whereas for the TU dataset collection it seems to offer no benefit. This might be due to the increased complexity of the data domains represented by the vision datasets (images) in comparison to the TU datasets (more traditional interactions of actors in a network of connections).
> In the newly added experiments in Section 4.5 we are able to gain a deeper analytical insight into the Expander GNN. We observe that the ExpanderGNN outperforms dense GNNs, when they have an equal number of parameters and hence, utilises the allocated parameters more efficiently. In another new series of experiments we are furthermore able to compare the Expander Linear Layer to a deterministic sampling mechanism with lower connectivity than the Expander LInear Layers. This enables us to conclude that also for GNNs the connectivity of the neural network architecture graph positively correlates with model performance.

---

> > ### Author Response · Authors · 2020-11-25
> > **Response to Reviewer 4: New dataset and theorectical analysis (2/2)**
> >
> > 5. *“The paper title is a bit misleading, I would suggest changing Analyzing to Simplifying, since the paper mainly proposes the method and shows on par performance, but the analysis is lacking.”*
> > This is a fair point of criticism, which we have considered carefully. We have come to the conclusion that our proposed model classes decompose the Update step present in GNNs into its different computational components and thereby enable an analysis of it. Furthermore, we have significantly extended the analysis of the results in Section 4. Therefore, we believe that the word “Analysing” is appropriately used in the title.
> >
> >
> > [1] Ameya Prabhu, Girish Varma & Anoop Namboodiri, “Deep expander networks:  Efficient deep networks from graph theory,” In: Proceedings of the European Conference on Computer Vision (ECCV), pp. 20 – 35, 2018.
> >
> > [2] Weihua Hu, Matthias Fey, Marinka Zitnik, Yuxiao Dong, Hongyu Ren, Bowen Liu, Michele Catasta & Jure Leskovec, "Open graph benchmark: Datasets for machine learning on graphs," arXiv preprint arXiv:2005.00687 (2020).
> >
> > [3] Vijay Prakash Dwivedi, Chaitanya K. Joshi, Thomas Laurent, Yoshua Bengio & Xavier Bresson, “Benchmarking graph neural networks,” arXiv:2003.00982, 2020.

---

### Official Review · AnonReviewer1 · 2020-10-28
**Interesting, but novelty and theoretical justification is limited**

**Rating:** 4
**Confidence:** 3

**Review:**

Analysing the update step in graph neural networks via sparsification

Summary:

The paper is focused on the Update step in a message passing (graph convolution) neural network. Several sparse variants of the update step, including its complete removal and expander graphs with varying density, are compared in a thorough and comprehensive empirical study.

Positive:

1. The paper is quite well written and easy to follow.
2. The research hypotheses are clear, and the experiments are well designed with that in mind.
3. Technical details are presented in sufficient mathematical detail.

Negative:

1. The authors do not appear to supply a software implementation, which would have been of great benefit to the reader.
2. The technical novelty/contribution of the paper is limited (main contribution is empirical)
3. The data sets in the study could perhaps be more fittingly chosen, so that the benefit of the graph neural network is more evident in all data sets. Not necessary to get the main message from the results, but would provide stronger motivation in my view.
4. The findings are not strongly justified by theory. Of course, not all work has to provide theoretical results, but a more technical discussion of the potential benefites of the motivation for particular Update structures could be interesting, and would strengthen the paper.

Recommendation:

Rejection. While the paper is well presented and technically sound, it would be much stronger if it could provide some theoretical analysis/justification of its findings.

Further comments:

The graph representation of MNIST and CIFAR10 was not clear to me.

---

> ### Author Response · Authors · 2020-11-25
> **Response to Reviewer 1: Further theorectical justification (1/2)**
>
> We would like to sincerely thank you for your constructive and helpful review. Following your comments we have submitted the software implementation for our comprehensive set of experiments, added further theoretical discussion in Section 3 and extended the discussion of the empirical results in Section 4. We have furthermore added three new sets of experiments to Section 4, which further motivate our proposed Update structure and enhance the analytical understanding of our results. We would now like to address your individual comments in the order in which you presented them.
>
>
> 1. We have added the code to the submission. Please find the zip file containing our implementation of the proposed models and performed experiments in the supplementary material. README.md contains the full set of instructions necessary to run our code and all required package dependencies.
>
>
> 2. We agree with the reviewer that the theoretical contributions presented in this paper are limited. Potentially the proposition of the Activation-Only model class is among the most significant theoretical advances we present. However, the focus of this paper lies in the study of an interesting problem, questioning the role of the Update step in GNNs. To gain an answer to this question we perform a thorough empirical study of significant size and impact. Our study covers a wide range of GNN models and different graph learning tasks and thus, enables us to draw a conclusion with great confidence. We therefore believe that this paper merits publication without a significant theoretical component.
>
>
> 3. We agree with the reviewer that the benefit of using graph neural networks is not visible for some of the datasets we have chosen. To slightly alleviate this shortcoming we have included new results on the large ogbn-arxiv dataset with 169,343 nodes and 1,166,243 edges, where the strength of GNN architectures is clearly visible.
> That being said, the fact that the benefits of GNNs are not clearly visible in all of our experiments has no impact on our conclusion. It is a known effect that on some of our graph datasets, such as ENZYMES and DD,  the simple MLP can achieve better performance than GNNs [1]. A recent study [2] has shown that more complex GNN models can outperform the MLP on the aforementioned datasets. However, the objective of the experiments is not to compare between different GNN models and find out the highest performing, but rather to compare within each GNN model, its simplified, sparsified and vanilla versions, in order to find out the importance of each component of the Update step. And our finding is unanimous across the majority of the chosen datasets: the linear transform in the Update step of GNNs can be sparsified to an arbitrary level and can even be omitted in many cases without a loss of performance.
>
>
> 4. Thank you very much for raising this point. In response we have expanded our discussion of the potential of expander graphs in the design of neural network architectures in Section 3.1.2. We now provide references to three papers which link the connectivity of the neural network architecture (which is high in expander graphs designs) to improved performance of neural networks [3,4] and the expressivity of the neural network structure [5]. This added theoretical discussion strengthens our case for the potential which expander graph neural network architectures have. In Section 4.5, we have furthermore added a new set of experiments, where we observe ourselves that the Expander GCN outperforms a GCN architecture sampled from a sparsifier, which has been shown to have a lower connectivity than our expander graph design. Therefore, we have significantly reinforced the motivation for our proposed Update structure and its theoretical discussion.
>
>
> 5. We thank the reviewer for pointing out the lack of clarity in how the graph representations are obtained from image data.  We have added an explanation of the conversion procedure from images to graphs for these two datasets to Appendix C.1. Specifically, we follow the procedure from [6], who begin by extracting small regions of homogeneous intensity from the images, named ``Superpixels'' [2], and then construct a $K-$nearest neighbour graph from these superpixels. For further technical details of the constructions, we refer the reviewer to [6].

---

> > ### Author Response · Authors · 2020-11-25
> > **Response to Reviewer 1: Further theorectical justification (2/2)**
> >
> > [1] Enxhell Luzhnica, Ben Day & Pietro Liò, “On Graph Classification Networks, Datasets and Baselines,” In: Workshop on Learning and Reasoning with Graph-Structured Data (ICML),  2019.
> >
> > [2]  Vijay Prakash Dwivedi, Chaitanya K. Joshi, Thomas Laurent, Yoshua Bengio & Xavier Bresson, “Benchmarking graph neural networks,” arXiv:2003.00982, 2020.
> >
> > [3]  Kepner, Jeremy, & Ryan Robinett. "Radix-net: Structured sparse matrices for deep neural networks." 2019 IEEE International Parallel and Distributed Processing Symposium Workshops (IPDPSW). IEEE, 2019.
> >
> > [4] Bourely, Alfred, John Patrick Boueri & Krzysztof Choromonski, "Sparse neural networks topologies," arXiv preprint arXiv:1706.05683 (2017).
> >
> > [5] Bölcskei, H., Grohs, P., Kutyniok, G., & Petersen, P., “Optimal approximation with sparsely connected deep neural networks." SIAM Journal on Mathematics of Data Science, 1(1), 8-45, 2019
> >
> > [6] Knyazev, Boris, Graham W. Taylor & Mohamed Amer, "Understanding attention and generalization in graph neural networks," Advances in Neural Information Processing Systems (NeurIPS), 2019.

---

### Official Review · AnonReviewer2 · 2020-10-29
**Some interesting empirical results but unclear impact**

**Rating:** 6
**Confidence:** 4

**Review:**

**Post response update:**
I would like to thank the authors for their response and revision addressing my concerns, at least to some extent. However, after careful consideration as well as looking at the other reviews (and responses), I am still not convinced the case can be made for increasing the score beyond marginal leaning towards accept, which was the original score I have marked for the paper. The paper does provide some interesting empirical evidence showing that GNNs can be simplified to by sparsifying, or even removing, the learned weights for combining different channels between message passing steps. However, this kind of simplification is not particular to GNNs (indeed, many popular network architectures are overparameterized and can be significantly sparsified), and has little effect on the incorporation of graph structure in the network. The activation-only experiments are perhaps more specific to GNNs, but on the other had, they mostly establish the importance of message passing, which has already been pretty well established in previous work (together with certain well described limitations of current designs). As other reviewers indicated, the insights provided here are intriguing, but it is not clear whether they meet the level of significance and impact expected from an ICLR publication.

---

Most popular graph neural networks (GNN) can be formulated as combining learned transformations between node features on graphs, parametrized by GNN weights, and nonlearned aggregation of information within neighborhoods determined by the graph structure. While most work on GNN focuses on how to improve the aggregation of information (for example, with message passing, convolutional graph filters, etc.), this work examines the redundancy in overparameterized network weights and shows it can often be significantly sparsified with relatively minor effect on network task performance, such as classification or regression accuracy. This, in turn, can lead to simpler models, with significantly less trained parameters, which in turn also leads to more efficient training process. The study here establishes this trend via extensive empirical study, using expander graph based sparsification of neural network weights (via modeling of neuron connections in each layer as a bipartite graph) following the sampling procedure proposed in Prabhu et al. (2018).

My main criticism of the paper is that it offers little insight specific to GNNs. Indeed, the so-called "Update" step considered here is essentially a way to mix together different channels of information at a node-wise level without considering graph structure, which is mainly used and encoded by the so-called "Aggregate" step in GNNs. As such, the sparsification here is not much different from the typical sparsification approaches applied to feed forward (or other) neural networks. It is generally a well accepted understanding in deep learning that neural networks are often unnecessarily overparameterized, and most of them can easily be pruned or sparsified (at least to some extent) without significantly degrading their task-specific results. Moreover, while several architectures are considered here (GCN, GIN, and standard MLP to provide a non-GNN baseline), it is not clear how well optimized the hyperparameters are for these networks prior to pruning. Are these networks as compact as they can be? or perhaps further tuning could also achieve a smaller network with roughly the same performance that doesn't need sparsification. Furthermore, the authors only consider a specific sparsification approach, but it seems to me that pretty much any sparsification or pruning approach could be applied here. How does the sparsification strategy affect the results? Perhaps a comparison of several sparsifiers could help establish the main argument of the paper, or justify the use of expander-graph sparsification, even though the graph here has nothing to do with the underlying data graph.

However, all that being said, the paper does provide evidence showing that with very few weights (or even no learned weights at all), GNNs are able to retain reasonable results, indicating some information is extracted solely by the message passing regardless of the trained network components. The discussion here is framed as following up on indications and ideas provided by Wu et al. (2019), and it is effective in this respect. Therefore, there may be some value in continuing to push the discussion along the research line here, to promote the study and better understanding of the impact different components have on the performance of GNNs. With this in mind, I consider this paper a borderline case, while slightly leaning towards acceptance in order to give the authors an opportunity to improve the presentation here by considering and studying the impact of sparsification strategies and presparsification hyperparameter tuning on presented results, in order to address my concerns mentioned above.

---

> ### Author Response · Authors · 2020-11-25
> **Response to Reviewer 2: Two new experiments (1/2)**
>
> We would like to sincerely thank you for your careful review. As a result of your comments we have been able to include two new experiments in the revised manuscript, where we compare the standard GCN with its ExpanderGCN counterpart and also assess the impact of different sparsification strategies.  In these comparisons we match the number of parameters of the compared models and observe that the ExpanderGCN outperforms the standard GCN as well as GCN architectures, sampled from samplers resulting in architectures with lower connectivity than the expander graphs.  As you suggested these additional experiments strengthen the motivation of the proposed models and help establish our overall conclusion.
>
> Please find below our more detailed responses to the main shortcomings you identified.
>
> > *"My main criticism of the paper is that it offers little insight specific to GNNs. [...] As such, the sparsification here is not much different from the typical sparsification approaches applied to feed forward (or other) neural networks.”*
>
> We agree that the minor theoretical contributions presented in the sparsification approach used in the ExpanderGNNs is not specific to GNNs. The sparsification approach has significant differences to conventional pruning approaches in that, following the work of Prabhu et al. [1], we initialise sparse structures rather than sparsifying trained architectures. The Activation-Only model class however, is a benchmark model which is only well defined in the context of GNNs and as such a contribution specific to GNNs.
>
> We believe that the main contribution of this paper, the comprehensive empirical study, does provide insight specific to GNN models. Our main finding in the extensive number of experiments is that in the majority of tasks and datasets the linear transformation in the Update step can be left out without harming the overall performance of the model.
>
> We find that in almost all experiments initialising the linear transformation layers with 10% of all possible parameters achieves equal results to models trained with 100% of parameters available to them. This empirical study gives us an understanding of the impact of the different components of the Update step in practice and is specific to GNNs. For conventional neural networks where single computational layers are not composed of an aggregation step and an update step similar levels of sparsification of the trained linear transform are unthinkable.
>
> >*“Moreover, while several architectures are considered here (GCN, GIN, and standard MLP to provide a non-GNN baseline), it is not clear how well optimized the hyperparameters are for these networks prior to pruning. Are these networks as compact as they can be?”*
>
> This is a very interesting question. Following your recommendation of further exploration of the hyperparameter space, we have added a set of experiments in the newly added Section 4.5, In these experiments we compare the GCN and the ExpanderGCN with a matching number of parameters, where the hidden dimension of the Update step was varied. We observe the ExpanderGCN to outperform the vanilla GCN for almost all hyperparameter settings. This new set of experiments allows us to demonstrate the efficient parameter usage the ExpanderGCN enables. Therefore, more compact GNN models are in fact outperformed by their ExpanderGNN counterpart.
>
> We furthermore want to clarify that in the existing series of experiments, in Sections 4.2, 4.3 and 4.4, we had chosen the hyperparameters of the models to match those determined in [2]. The topic in [2] is to provide a fair and informative comparison framework for GNN models by optimising the hyperparameters of each model to produce the best possible performance. Therefore, we believe it makes the experiments in this paper more easily comparable with results obtained in the literature and thereby more relevant if we adhere to the hyperparameters determined in [2]. We agree that this design choice should have been discussed in the main body of the text and we have therefore added it to Section 4.1.

---

> > ### Author Response · Authors · 2020-11-25
> > **Response to Reviewer 2: Two new experiments (2/2)**
> >
> > >*“Furthermore, the authors only consider a specific sparsification approach, but it seems to me that pretty much any sparsification or pruning approach could be applied here. How does the sparsification strategy affect the results?”*
> >
> > Thank you very much for suggesting this study. We have added an experiment to the newly added Section 4.5, where we compare the performance of the Expander GCN and a GCN where the Expander Linear Layer is replaced by deterministic graphs obtained from a sampler published in [3], which is known to produce graphs with lower connectivity than the Expander Linear Layer. We observe that the Expander GCN outperforms this deterministic architecture for almost all parameter configurations. This set of experiments serves as further evidence motivating the Expander GNN and also  confirms the positive correlation of the connectivity of the neural network architecture and the performance of the neural network.
> >
> > [1] Ameya Prabhu, Girish Varma & Anoop Namboodiri, “Deep expander networks:  Efficient deep networks from graph theory,” In: Proceedings of the European Conference on Computer Vision (ECCV), pp. 20 – 35, 2018.
> >
> > [2] Vijay Prakash Dwivedi, Chaitanya K. Joshi, Thomas Laurent, Yoshua Bengio & Xavier Bresson, “Benchmarking graph neural networks,” arXiv:2003.00982, 2020.
> >
> > [3] Bourely, Alfred, John Patrick Boueri & Krzysztof Choromonski. "Sparse neural networks topologies." arXiv preprint arXiv:1706.05683 (2017).

---

### Decision · Program_Chairs · 2021-01-07
**Final Decision**

**Decision:**

Reject

**Comment:**

The paper focuses on the update step in Message-Passing Neural Networks, specifically for GNN. A series of sparse variants of the update step, say complete removal and expander graphs with varying density, are compared in empirical studies. The findings are quite useful for practice, and the paper is organized and written well.  As observed by the reviewers, there are several concerns regarding the novelty and contribution of the work. Besides, theoretical analysis of the sparsification approach is lacking. The authors provided a good rebuttal and addressed some concerns, but not to the degree that reviewers think it passes the bar of ICLR. We encourage the authors to further improve the work to address the key concerns.